# Error Analysis of Spherically Constrained Least Squares Reformulation in Solving the Stackelberg Prediction Game

**Xiyuan Li**    **Weiwei Liu**[*]
School of Computer Science, Wuhan University
National Engineering Research Center for Multimedia Software, Wuhan University
Institute of Artificial Intelligence, Wuhan University
Hubei Key Laboratory of Multimedia and Network Communication Engineering, Wuhan University
Lee_xiyuan@outlook.com, liuweiwei863@gmail.com

## Abstract

The Stackelberg prediction game (SPG) is a popular model for characterizing strategic interactions between a learner and an adversarial data provider. Although optimization problems in SPGs are often NP-hard, a notable special case involving the least squares loss (SPG-LS) has gained significant research attention recently [1, 2, 3]. The latest state-of-the-art method for solving the SPG-LS problem is the spherically constrained least squares reformulation (SCLS) method proposed in the work of [3]. However, the paper [3] lacks theoretical analysis on the error of the SCLS method, which limits its large-scale applications. In this paper, we investigate the estimation error between the learner obtained by the SCLS method and the actual learner. Specifically, we reframe the estimation error of the SCLS method as a Primary Optimization (**PO**) problem and utilize the Convex Gaussian min-max theorem (CGMT) to transform the **PO** problem into an Auxiliary Optimization (**AO**) problem. Subsequently, we provide a theoretical error analysis for the SCLS method based on this simplified **AO** problem. This analysis not only strengthens the theoretical framework of the SCLS method but also confirms the reliability of the learner produced by it. We further conduct experiments to validate our theorems, and the results are in excellent agreement with our theoretical predictions.

## 1   Introduction

The Stackelberg prediction games (SPGs) play prominent roles in various applications of the machine learning field, such as intrusion detection [4], spam filtering [5], and malware detection [6, 7]. SPG characterizes the interactions between two players, a learner and a data provider (attacker), during the training process of various machine learning algorithms [8, 9, 10, 11]. Specifically, the learner first selects a learning model to fit the given data. The data provider, with full knowledge of the learner's model, then attacks the learner by modifying the data. The learner's goal is to minimize its loss function under the assumption that the training data has been optimally modified from the data provider's perspective. Therefore, the SPG model is often formulated as a bi-level optimization problem, which is generally NP-hard even in the simplest case with linear constraints and objectives [1, 12].

To overcome the NP-hard nature of SPGs, [1, 2, 3] focus on a commonly used subclass of SPGs, termed as the SPG-LS, whose loss functions for the learner and the data provider are least squares. Specifically, SPG-LS has access to a set of $n$ sample tuples denoted by $S = \{(\boldsymbol{x}_i, y_i, z_i)\}_{i=1}^n$, where

---

[*]Corresponding author: Weiwei Liu (liuweiwei863@gmail.com).

38th Conference on Neural Information Processing Systems (NeurIPS 2024).

$\boldsymbol{x}_i \in \mathbb{R}^d$ is input data with $d$ features, $y_i$ is the true output label of $\boldsymbol{x}_i$, and $z_i$ is the label that the data provider aims to achieve. The learner of SPG-LS aims to train a linear predictor $\boldsymbol{w} \in \mathbb{R}^d$ to best estimate the true output label $y_i$ of the fake data $\boldsymbol{x}_i^*$ by minimizing the least squares loss:

$$\boldsymbol{w}^* = \arg\min_{\boldsymbol{w}} \frac{1}{n} \sum_{i=1}^{n} \|\boldsymbol{w}^\top \boldsymbol{x}_i^* - y_i\|^2.$$

Meanwhile, the data provider of SPG-LS, with full knowledge of the learner's predictive model $\boldsymbol{w}$, selects the following least squares attacking strategy (i.e., modifying the data $\hat{\boldsymbol{x}}_i$) to make the corresponding prediction $\boldsymbol{w}^\top \boldsymbol{x}_i^*$ close to the desired label $z_i$:

$$\boldsymbol{x}_i^* = \arg\min_{\hat{\boldsymbol{x}}} \|\boldsymbol{w}^\top \hat{\boldsymbol{x}}_i - z_i\|^2 + \gamma \|\boldsymbol{x}_i - \hat{\boldsymbol{x}}_i\|^2,$$

where $\gamma > 0$ is a regularizer to adjust the trade-off between the deviation from the original data $\boldsymbol{x}_i$ and closeness to the target $z_i$. Thus, the SPG-LS model can be expressed as the following bi-level optimization problem, as described in [1, 2, 3]:

$$\min_{\boldsymbol{w}} \|\boldsymbol{X}^* \boldsymbol{w} - \boldsymbol{y}\|^2, \quad \text{s.t. } \boldsymbol{X}^* = \arg\min_{\hat{\boldsymbol{X}}} \|\hat{\boldsymbol{X}} \boldsymbol{w} - \boldsymbol{z}\|^2 + \gamma \|\hat{\boldsymbol{X}} - \boldsymbol{X}\|_F^2, \quad (1)$$

where $\boldsymbol{X} = (\boldsymbol{x}_1, \boldsymbol{x}_2, \cdots, \boldsymbol{x}_n)^\top \in \mathbb{R}^{n \times d}$ is the input sample matrix, $\boldsymbol{y} = (y_1, y_2, \cdots, y_n)^\top \in \mathbb{R}^n$ is the vector of true output labels, and $\boldsymbol{z} = (z_1, z_2, \cdots, z_n)^\top \in \mathbb{R}^n$ is the vector of labels that the attacker aims to achieve. Moreover, $\|\cdot\|$ denotes the Euclidean norm ($l_2$) unless otherwise specified.

There have been several studies solving the SPG-LS (1) to determine the Stackelberg equilibrium point between the learner and the data provider. The initial step in reformulating SPG-LS (1) is taken by [1], who provides a single-level quadratic fractional program (QFP) that can be globally solved by a bisection algorithm. However, this QFP method is computationally prohibitive in practice due to the need to solve multiple semidefinite programs (SDPs). Later, [2] improves upon [1] by showing that the SPG-LS (1) can be globally solved by reducing a single SDP to a second-order cone program (SOCP). Despite being faster than the QFP method, the SOCP method is still not well-suited for large-scale SPG-LS (1) due to time-consuming spectral decomposition. Recently, [3] proposes a spherically constrained least squares reformulation (SCLS) method, addressing the above-mentioned issues with a novel nonlinear change of variables. Furthermore, [3] demonstrates that the SCLS method outperforms the SOCP method and is currently the state-of-the-art for solving SPG-LS (1), having won the ICML 2022 Outstanding Paper Award.

However, the lack of theoretical analysis on the error of the SCLS method limits its large-scale practical applications. In this paper, we investigate the estimation error between the learner (e.t.$\boldsymbol{w}^*$) estimated by the SCLS method and the true learner (denoted as $\boldsymbol{w}_0$) to validate the reliability of $\boldsymbol{w}^*$. Specifically, we assume the samples $S = \{(\boldsymbol{x}_i, y_i, z_i)\}_{i=1}^n$ are generated by the following black box model:

$$\boldsymbol{X}^* = \arg\min_{\hat{\boldsymbol{X}}} \|\hat{\boldsymbol{X}} \boldsymbol{w}_0 - \boldsymbol{z}\|^2 + \gamma \|\hat{\boldsymbol{X}} - \boldsymbol{X}\|_F^2, \quad \boldsymbol{y} = \boldsymbol{X}^* \boldsymbol{w}_0 + \boldsymbol{\epsilon}, \quad (2)$$

where $\boldsymbol{w}_0 \in \mathbb{R}^d$ represents the "true" weight parameter of the real learner, and $\boldsymbol{\epsilon} = (\epsilon_1, \epsilon_2, \cdots, \epsilon_n)^\top \in \mathbb{R}^n$ is the noise vector. Moreover, the entries of $\boldsymbol{X}$ and $\boldsymbol{z}$ are drawn i.i.d. from $\mathcal{N}(0, 1)$; the entries of $\boldsymbol{\epsilon}$ are drawn i.i.d. from $\mathcal{N}(0, \sigma^2)$; and we assume $\lim_{n \to \infty} \frac{d}{n} \in (0, 1)$. Given $\boldsymbol{X}, \boldsymbol{z}$, and $\boldsymbol{y}$ generated by this model (2), we solve SPG-LS (1) by the SCLS method to obtain $\boldsymbol{w}^*$ that is used to estimate the target vector $\boldsymbol{w}_0$. Our task is to measure the optimal estimation error of the SCLS method, represented by $\|\boldsymbol{w}^* - \boldsymbol{w}_0\|$.

We start by formulating an optimization problem regarding the estimation error (e.t. $\boldsymbol{\beta} := \boldsymbol{w} - \boldsymbol{w}_0$) of the SCLS method. Subsequently, we convert this optimization problem into a Primary Optimization (**PO**) problem, employing the Fenchel-Moreau theorem [13]. Following this, we utilize the Convex Gaussian min-max theorem (CGMT) to simplify the **PO** problem into an Auxiliary Optimization (**AO**) problem. Finally, we conduct a theoretical error analysis of the SCLS method based on the **AO** problem. Our main theoretical result can be summarized as follows:

$$\lim_{n \to \infty} \|\boldsymbol{w}^* - \boldsymbol{w}_0\| \xrightarrow{P} 0, \quad (3)$$

which guarantees the reliability of $\boldsymbol{w}^*$ learned by the SCLS method. Our analysis strengthens the theoretical foundations of the SCLS method and provides theoretical support for its broad applications.

We also conduct experiments to validate our theorems. The results show that, as $n$ goes to $\infty$, the parameter vector $\boldsymbol{w}^*$ learned through the SCLS method converges to actual parameter vector $\boldsymbol{w}_0$ in probability, which aligns excellently with our theoretical predictions.

## 1.1 Outline

The structure of the remaining sections in this paper is organized as follows: Additional related work is discussed in Appendix 2. Section 3 provides an overview of the SCLS method, CGMT technology, and foundational concepts. Our main inference processes are detailed in Section 4. Specifically, in Section 4.1, we present an optimization problem concerning the estimation error of the approximated SCLS method and establish the relationship between the original and the approximated SCLS method; In Section 4.2, the estimation error of the approximated SCLS method is transformed into a **PO** problem; In Section 4.3, we simplify the **PO** problem to an **AO** problem using CGMT; In Section 4.4, we conduct an estimation error analysis of the SCLS method based on the **AO** problem and the relationship between the original and the approximated SCLS method. We then present the experimental results in Section 5. Finally, a summary is provided in Section 6. Moreover, the limitations of our work are detailed in Appendix A

## 2 Related Work

### 2.1 The Stackelberg prediction game

Stackelberg Prediction Games (SPGs) were initially introduced by [14], drawing inspiration from Stackelberg competition—a model initially developed to describe market behaviors. A notable parallel can be drawn with Stackelberg Security Games (SSGs), as detailed by [4, 6]. In SSGs, a defender strategically allocates resources to protect targets from an attacker. The optimal defense strategy in SSGs typically involves solving multiple linear programs [15, 16], and [17] demonstrates that a near-optimal strategy can be efficiently approximated through a polynomial number of queries to the attacker's model. While both SPG and SSG frame the learning of an optimal strategy as a bilevel optimization problem, SPGs are distinctly designed to counteract manipulation within machine learning algorithms [1].

Recognized as NP-hard hierarchical mathematical challenges [12], SPGs have been extensively examined. However, existing research predominantly addresses scenarios where data providers exhibit partial adversarial behaviors or possess constrained adversarial capabilities. The study by [1] explores SPGs within the context of linear least squares regression (e.g., SPG-LS), assuming data providers are neither fully adversarial nor entirely honest. To address the SPG-LS problem, [1] initially formulated a solution approach, which was later enhanced by [2]. Subsequently, [3] introduced the Spherical Constrained Least Squares (SCLS) method, currently acknowledged as the state-of-the-art. This paper aims to further elucidate the error dynamics of the SCLS method [18], thereby solidifying its theoretical foundation for broader practical application.

### 2.2 The Gaussian Min-max Theorem

The Convex Gaussian Min-max Theorem (CGMT) framework, first introduced by [19], serves as a potent analytical tool extensively utilized to evaluate the performance of solutions within non-smooth regularized convex optimization problems. This framework is derived from Gordon's Gaussian Min-max Theorem (GMT) [20, 21], which provides foundational insights into the behavior of Gaussian processes in optimization scenarios. Over the years, the CGMT has enabled significant advancements across a spectrum of practical applications. Notable examples include enhancements in regularized logistic regression [22], max-margin classifiers [23] and adversarial training [24, 25]. Motivated by these successful applications, we are encouraged to employ the CGMT framework to conduct a thorough analysis of the estimation error associated with the SCLS method.

# 3 Preliminaries

## 3.1 The SCLS method

This section provides a comprehensive overview of the SCLS method as introduced by [3]. Expanding upon previous studies by [1, 2], [3] reformulates SPG-LS (1) into the following quadratic fractional program (QFP) utilizing the Sherman-Morrison formula [26]:

$$\inf_{\boldsymbol{w}} \left\| \frac{\frac{1}{\gamma} \boldsymbol{z} \boldsymbol{w}^\top \boldsymbol{w} + \boldsymbol{X} \boldsymbol{w}}{1 + \frac{1}{\gamma} \boldsymbol{w}^\top \boldsymbol{w}} - \boldsymbol{y} \right\|^2 . \tag{4}$$

Moreover, [3] introduces an augmented variable $\alpha = \boldsymbol{w}^\top \boldsymbol{w}/\gamma$ and further reformulate QFP (4) as:

$$\inf_{\boldsymbol{w},\alpha} v(\boldsymbol{w},\alpha) \triangleq \left\| \frac{\alpha \boldsymbol{z} + \boldsymbol{X} \boldsymbol{w}}{1 + \alpha} - \boldsymbol{y} \right\|^2, \quad \text{s.t. } \boldsymbol{w}^\top \boldsymbol{w} = \gamma \alpha. \tag{5}$$

Subsequently, [3] makes a assumption on the nonemptiness of the optimal solution set of QFP (5).

**Assumption 3.1** ([3]). Assume that the optimal solution set of (5) (or equivalently, (4)) is nonempty.

Under Assumption 3.1, [3] employs a nonlinear variable transformation to recast the QFP (5) as a spherical constrained least squares (SCLS) problem:

$$\min_{\tilde{\boldsymbol{w}},\tilde{\alpha}} \quad \tilde{v}(\tilde{\boldsymbol{w}},\tilde{\alpha}) \triangleq \left\| \frac{\tilde{\alpha}}{2} \boldsymbol{z} + \frac{\sqrt{\gamma}}{2} \boldsymbol{X} \tilde{\boldsymbol{w}} - (\boldsymbol{y} - \frac{\boldsymbol{z}}{2}) \right\|^2, \quad \text{s.t. } \tilde{\boldsymbol{w}}^\top \tilde{\boldsymbol{w}} + \tilde{\alpha}^2 = 1, \tag{6}$$

where $\tilde{\boldsymbol{w}}$ and $\tilde{\alpha}$ are defined in Lemmas 3.2 and 3.3. Upon identifying a feasible solution in QFP (5), [3] introduces Lemma 3.2 to construct a feasible solution in SCLS (6), while Lemma 3.3 describes the inverse transformation.

**Lemma 3.2** ([3]). *Suppose* $(\boldsymbol{w},\alpha)$ *is a feasible solution of QFP (5). Then* $(\tilde{\boldsymbol{w}},\tilde{\alpha})$, *defined as*

$$\tilde{\boldsymbol{w}} := \frac{2}{\sqrt{\gamma}(\alpha+1)} \boldsymbol{w} \quad \text{and} \quad \tilde{\alpha} := \frac{\alpha-1}{\alpha+1}, \tag{7}$$

*is feasible to SCLS (6) and* $v(\boldsymbol{w},\alpha) = \tilde{v}(\tilde{\boldsymbol{w}},\tilde{\alpha})$.

**Lemma 3.3** ([3]). *Suppose* $(\tilde{\boldsymbol{w}},\tilde{\alpha})$ *is feasible to SCLS (6) with* $\tilde{\alpha} \neq 1$. *Then* $(\boldsymbol{w},\alpha)$, *defined as*

$$\boldsymbol{w} := \frac{\sqrt{\gamma}}{1-\tilde{\alpha}} \tilde{\boldsymbol{w}} \quad \text{and} \quad \alpha := \frac{1+\tilde{\alpha}}{1-\tilde{\alpha}}, \tag{8}$$

*is feasible to QFP (5) and* $\tilde{v}(\tilde{\boldsymbol{w}},\tilde{\alpha}) = v(\boldsymbol{w},\alpha)$.

Let $v^*$ and $\tilde{v}^*$ represent the optimal values of QFP (5) and SCLS (6), respectively. Subsequently, [3] presents Theorem 3.4 to elucidate the relationship between the solutions of QFP (5) and SCLS (6).

**Theorem 3.4** ([3]). *Given Assumption 3.1, then there exists an optimal solution* $(\tilde{\boldsymbol{w}},\tilde{\alpha})$ *to SCLS (6) with* $\tilde{\alpha} \neq 1$. *Moreover,* $(\boldsymbol{w},\alpha)$, *defined by (8), is an optimal solution to (5) and* $v^* = v(\boldsymbol{w},\alpha) = \tilde{v}(\tilde{\boldsymbol{w}},\tilde{\alpha}) = \tilde{v}^*$.

Theorem 3.4 indicates that an optimal solution of SCLS (6) can be utilized to recover an optimal solution of SPG-LS (1). Additionally, the converse of this theorem also holds, as demonstrated by [3]. Specifically, if $(\boldsymbol{w},\alpha)$ is an optimal solution to QFP (5), $(\tilde{\boldsymbol{w}},\tilde{\alpha})$, as defined by (7), is also an optimal solution to SCLS (6).

It is important to note that [3] focus on a compact form of SCLS (6):

$$\min_{\boldsymbol{r}} \ q(\boldsymbol{r}), \quad \text{s.t. } \boldsymbol{r}^T \boldsymbol{r} = 1, \tag{9}$$

where $q(\boldsymbol{r}) = \|\hat{L}\boldsymbol{r} - (\boldsymbol{y} - \boldsymbol{z}/2)\|^2$, $\hat{L} = \left( \frac{\sqrt{\gamma}}{2} \boldsymbol{X} \quad \frac{\boldsymbol{z}}{2} \right)$ and $\boldsymbol{r} = \begin{pmatrix} \tilde{\boldsymbol{w}} \\ \tilde{\alpha} \end{pmatrix}$. Due to the equivalence of SCLS problems (6) and (9), this paper focus on the SCLS (6).

## 3.2 The Convex Gaussian Min-max Theorem

The Convex Gaussian Min-max Theorem (CGMT) originates from Gordon's Gaussian Min-max Theorem (GMT) [21], which provides probabilistic bounds on the optimal cost of **PO** problem via a simpler **AO** problem. CGMT further tightens the bounds under convexity assumptions. Building on GMT, [20] introduces the following asymptotic sequence and notation.

**Definition 3.5** (GMT admissible sequence). The sequence $\big\{\boldsymbol{G}^{(d)}, \boldsymbol{g}^{(d)}, \boldsymbol{h}^{(d)}, \mathcal{S}_{\boldsymbol{\beta}}^{(d)}, \mathcal{S}_{\boldsymbol{u}}^{(d)}, \psi^{(d)}\big\}_{d \in \mathbb{N}}$ indexed by $d$, with $\boldsymbol{G}^{(d)} \in \mathbb{R}^{n \times d}$, $\boldsymbol{g}^{(d)} \in \mathbb{R}^n$, $\boldsymbol{h}^{(d)} \in \mathbb{R}^d$, $\mathcal{S}_{\boldsymbol{\beta}}^{(d)} \subset \mathbb{R}^d$, $\mathcal{S}_{\boldsymbol{u}}^{(d)} \subset \mathbb{R}^n$, $\psi^{(d)} : \mathcal{S}_{\boldsymbol{\beta}}^{(d)} \times \mathcal{S}_{\boldsymbol{u}}^{(d)} \to \mathbb{R}$ and $n = n(d)$, is said to be admissible if, for each $d \in \mathbb{N}$, $\mathcal{S}_{\boldsymbol{\beta}}^{(d)}$ and $\mathcal{S}_{\boldsymbol{u}}^{(d)}$ are compact sets and $\psi^{(d)}$ is continuous on its domain. Onwards, we will drop the superscript $(d)$ from $\boldsymbol{G}^{(d)}$, $\boldsymbol{g}^{(d)}$, $\boldsymbol{h}^{(d)}$.

A sequence $\big\{\boldsymbol{G}^{(d)}, \boldsymbol{g}^{(d)}, \boldsymbol{h}^{(d)}, \mathcal{S}_{\boldsymbol{\beta}}^{(d)}, \mathcal{S}_{\boldsymbol{u}}^{(d)}, \psi^{(d)}\big\}_{d \in \mathbb{N}}$ defines a sequence of min-max problems

$$\Phi^{(d)}(\boldsymbol{G}) := \min_{\boldsymbol{\beta} \in \mathcal{S}_{\boldsymbol{\beta}}^{(d)}} \max_{\boldsymbol{u} \in \mathcal{S}_{\boldsymbol{u}}^{(d)}} \boldsymbol{u}^\top \boldsymbol{G} \boldsymbol{\beta} + \psi^{(d)}(\boldsymbol{\beta}, \boldsymbol{u}) \tag{10}$$

$$\phi^{(d)}(\boldsymbol{g}, \boldsymbol{h}) := \min_{\boldsymbol{\beta} \in \mathcal{S}_{\boldsymbol{\beta}}^{(d)}} \max_{\boldsymbol{u} \in \mathcal{S}_{\boldsymbol{u}}^{(d)}} \|\boldsymbol{\beta}\| \boldsymbol{g}^\top \boldsymbol{u} + \|\boldsymbol{u}\| \boldsymbol{h}^\top \boldsymbol{\beta} + \psi^{(d)}(\boldsymbol{\beta}, \boldsymbol{u}) \tag{11}$$

Importantly, the formulation (10) is called Primary Optimization (**PO**) and the formulation (11) is called Auxiliary Optimization (**AO**). Additionally, let $\boldsymbol{\beta}_{\Phi}^{(d)}(\boldsymbol{G})$ denote the optimal minimizer of the **PO** problem (10), and $\boldsymbol{\beta}_{\phi}^{(d)}(\boldsymbol{g}, \boldsymbol{h})$ denote the optimal minimizer of the **AO** problem (11). Define $\upsilon^{(d)} : \mathcal{S}_{\boldsymbol{\beta}}^{(d)} \to \mathbb{R}$ as follows,

$$\upsilon^{(d)}(\boldsymbol{\beta}; \boldsymbol{g}, \boldsymbol{h}) := \max_{\boldsymbol{u} \in \mathcal{S}_{\boldsymbol{u}}^{(d)}} \|\boldsymbol{\beta}\| \boldsymbol{g}^\top \boldsymbol{u} + \|\boldsymbol{u}\| \boldsymbol{h}^\top \boldsymbol{\beta} + \psi^{(d)}(\boldsymbol{\beta}, \boldsymbol{u}). \tag{12}$$

Clearly, $\phi^{(d)}(\boldsymbol{g}, \boldsymbol{h}) := \min_{\boldsymbol{\beta} \in \mathcal{S}_{\boldsymbol{\beta}}^{(d)}} \upsilon^{(d)}(\boldsymbol{\beta}; \boldsymbol{g}, \boldsymbol{h})$. For a sequence of random variables $\{\mathcal{X}^{(d)}\}_{d \in \mathbb{N}}$ and a constant $c \in \mathbb{R}$, $\mathcal{X}^{(d)} \xrightarrow{P} c$ denotes convergence in probability, i.e., $\forall \epsilon > 0, \lim_{d \to \infty} \mathbb{P}\big(|\mathcal{X}^{(d)} - c| > \epsilon\big) = 0$. Based on the GMT admissible sequence and the notation introduced above, we present the CGMT below.

**Theorem 3.6** (CGMT [27]). *Let $\big\{\boldsymbol{G}^{(d)}, \boldsymbol{g}^{(d)}, \boldsymbol{h}^{(d)}, \mathcal{S}_{\boldsymbol{\beta}}^{(d)}, \mathcal{S}_{\boldsymbol{u}}^{(d)}, \psi^{(d)}\big\}_{d \in \mathbb{N}}$ be a GMT admissible sequence as in Definition 3.5, for which additionally the entries of $\boldsymbol{G}$, $\boldsymbol{g}$, $\boldsymbol{h}$ are drawn i.i.d. from $\mathcal{N}(0, 1)$. Let $\Phi^{(d)}(\boldsymbol{G})$, $\phi^{(d)}(\boldsymbol{g}, \boldsymbol{h})$ be the optimal costs, and, $\boldsymbol{\beta}_{\Phi}^{(d)}(\boldsymbol{G})$, $\boldsymbol{\beta}_{\phi}^{(d)}(\boldsymbol{g}, \boldsymbol{h})$ the corresponding optimal minimizers of the PO and AO problems in (10) and (11). The following three statements hold*

  *(i) For any $d \in \mathbb{N}$ and $c \in \mathbb{R}$, $\mathbb{P}\big(\Phi^{(d)}(\boldsymbol{G}) < c\big) \leq 2\mathbb{P}\big(\phi^{(d)}(\boldsymbol{g}, \boldsymbol{h}) \leq c\big)$.*

  *(ii) For any $d \in \mathbb{N}$. If $\mathcal{S}_{\boldsymbol{\beta}}^{(d)}$, $\mathcal{S}_{\boldsymbol{u}}^{(d)}$ are convex, and, $\psi^{(d)}(\cdot, \cdot)$ is convex-concave on $\mathcal{S}_{\boldsymbol{\beta}}^{(d)} \times \mathcal{S}_{\boldsymbol{u}}^{(d)}$, then, for any $\mu \in \mathbb{R}$ and $t > 0$,*

$$\mathbb{P}\big(|\Phi^{(d)}(\boldsymbol{G}) - \mu| > t\big) \leq 2\mathbb{P}\big(|\phi^{(d)}(\boldsymbol{g}, \boldsymbol{h}) - \mu| > t\big)$$

  *(iii) Assume the conditions of (ii) hold for all $d \in \mathbb{N}$. Let $\|\cdot\|$ denote some norm in $\mathbb{R}^d$ and recall (12). If, there exist constants (independent of $d$) $\kappa^*$, $\rho^*$ and $\tau > 0$ such that*

  *(a) $\phi^{(d)}(\boldsymbol{g}, \boldsymbol{h}) \xrightarrow{P} \kappa^*$, (b) $\|\boldsymbol{\beta}_{\phi}^{(d)}(\boldsymbol{g}, \boldsymbol{h})\| \xrightarrow{P} \rho^*$, (c) with probability one in the limit $d \to \infty$*

$$\Big\{\upsilon^{(d)}(\boldsymbol{\beta}; \boldsymbol{g}, \boldsymbol{h}) \geq \phi^{(d)}(\boldsymbol{g}, \boldsymbol{h}) + \tau\big(\|\boldsymbol{\beta}\| - \boldsymbol{\beta}_{\phi}^{(d)}(\boldsymbol{g}, \boldsymbol{h})\big)^2, \forall \boldsymbol{\beta} \in \mathcal{S}_{\boldsymbol{\beta}}^{(d)}\Big\},$$

  *then,*

$$\|\boldsymbol{\beta}_{\Phi}^{(d)}(\boldsymbol{G})\| \xrightarrow{P} \rho^*.$$

Theorem 3.6 indicates that, if the optimal cost $\phi(\boldsymbol{g}, \boldsymbol{h})$ of (11) converges to some value $\mu$, the same holds true for $\Phi(\boldsymbol{G})$ of **PO** (10). Furthermore, under appropriate additional assumptions, the optimal solutions of the **AO** and **PO** problems are also closely related by

$$\|\boldsymbol{\beta}_{\Phi}(\boldsymbol{G})\| = \|\boldsymbol{\beta}_{\phi}(\boldsymbol{g}, \boldsymbol{h})\|.$$

This suggests that within the CGMT framework, a challenging **PO** problem can be replaced with a simplified **AO** problem, from which the optimal solution of the **PO** problem can be accurately inferred [27]. Subsequently, we rewrite the estimation error of the SCLS method (6) in the form of **PO** problem (10) and analyze the minimizer of the simplified **AO** problem instead.

### 3.3 Basic Concept

**Conjugate pairs:** Consider a function $f : \mathbb{R}^d \to \mathbb{R}$. The Fenchel conjugate of $f$, denoted by $f^*$, is defined as $f^*(\boldsymbol{u}) = \sup_{\boldsymbol{v}} \boldsymbol{v}^\top \boldsymbol{u} - f(\boldsymbol{v})$, which is always convex and lower semi-continuous. By the Fenchel-Moreau theorem [13], if $f$ is both convex and continuous, then $f(\boldsymbol{v}) = \sup_{\boldsymbol{u}} \boldsymbol{u}^\top \boldsymbol{v} - f^*(\boldsymbol{u})$. In this paper, we consider the following conjugate pairs for the $l_2$ norm:

$$f(\boldsymbol{v}) = \|\boldsymbol{v}\|^2 \leftrightarrow f^*(\boldsymbol{u}) = \frac{\|\boldsymbol{u}\|^2}{4}. \tag{13}$$

**First-order approximation:** Assume $f$ is differentiable. According to [13, Theorem 23.4]:

$$f(\boldsymbol{w}) = f(\boldsymbol{w}_0) + [\dot{f}(\boldsymbol{w}_0)]^\top \boldsymbol{\beta} + O(\|\boldsymbol{\beta}\|^2) \tag{14}$$

where $\boldsymbol{w} = \boldsymbol{w}_0 + \boldsymbol{\beta}$ and $\dot{f}(\boldsymbol{w}_0) = \frac{\partial f}{\partial \boldsymbol{w}}\big|_{\boldsymbol{w}=\boldsymbol{w}_0}$. The linearization of $f(\cdot)$ around the interest $\boldsymbol{w}_0$ is

$$\hat{f}(\boldsymbol{w}) = f(\boldsymbol{w}_0) + [\dot{f}(\boldsymbol{w}_0)]^\top \boldsymbol{\beta}. \tag{15}$$

As $\|\boldsymbol{\beta}\|$ approaches 0, $\hat{f}(\boldsymbol{w})$ closely approximates $f(\boldsymbol{w})$.

## 4 The Error Analysis for the SCLS method

### 4.1 From the SCLS Method to PO

Given that the sample $(\boldsymbol{X}, \boldsymbol{y}, \boldsymbol{z})$ is generated by black box model (2), we integrate SPG-LS (1) and QFP (4) as follows:

$$\boldsymbol{y} = \boldsymbol{X}^* \boldsymbol{w}_0 + \boldsymbol{\epsilon} = \frac{\alpha_0 \boldsymbol{z} + \boldsymbol{X} \boldsymbol{w}_0}{1 + \alpha_0} + \boldsymbol{\epsilon}, \tag{16}$$

where $\alpha_0 = \boldsymbol{w}_0^\top \boldsymbol{w}_0 / \gamma$. Drawing inspiration from Lemmas 3.2 and 3.3, we define

$$\tilde{\boldsymbol{w}}_0 := \frac{2}{\sqrt{\gamma}(\alpha_0 + 1)} \boldsymbol{w}_0, \tilde{\alpha}_0 := \frac{\alpha_0 - 1}{\alpha_0 + 1}, \text{ with the inverses: } \boldsymbol{w}_0 := \frac{\sqrt{\gamma}}{1 - \tilde{\alpha}_0} \tilde{\boldsymbol{w}}_0, \alpha_0 := \frac{1 + \tilde{\alpha}_0}{1 - \tilde{\alpha}_0}. \tag{17}$$

This representation of $\tilde{\boldsymbol{w}}_0$ denotes the true weight parameter of SCLS (6). Notably, $(\tilde{\boldsymbol{w}}_0, \tilde{\alpha}_0)$ is valid as long as $\alpha_0 \geq 0$ and $\tilde{\alpha}_0 \neq 1$, which conforms to $(\tilde{\boldsymbol{w}}, \tilde{\alpha})$ defined by [3]. Specifically, For $\alpha_0 = \frac{\boldsymbol{w}_0^\top \boldsymbol{w}_0}{\gamma} \in (1, +\infty)$, $\tilde{\alpha}_0 > 0$; For $\alpha_0 = \frac{\boldsymbol{w}_0^\top \boldsymbol{w}_0}{\gamma} \in (0, 1)$, $\tilde{\alpha}_0 < 0$. Additionally, $(\tilde{\boldsymbol{w}}_0, \tilde{\alpha}_0)$ satisfies the constraint of SCLS (6), due to

$$\tilde{\boldsymbol{w}}_0^\top \tilde{\boldsymbol{w}}_0 + \tilde{\alpha}_0^2 = \frac{4}{\gamma(\alpha_0 + 1)^2} \boldsymbol{w}_0^\top \boldsymbol{w}_0 + \frac{(\alpha_0 - 1)^2}{(\alpha_0 + 1)^2} = \frac{4\alpha_0 + (\alpha_0 - 1)^2}{(\alpha_0 + 1)^2} = 1. \tag{18}$$

Taking (17) into (16), the expression of $\boldsymbol{y}$ can be rewritten as:

$$\begin{aligned}
\boldsymbol{y} &= \frac{\alpha_0 \boldsymbol{z} + \boldsymbol{X} \boldsymbol{w}_0}{\alpha_0 + 1} + \boldsymbol{\epsilon} = \frac{\alpha_0 - 1}{2(\alpha_0 + 1)} \boldsymbol{z} + \frac{\sqrt{\gamma}}{2} \boldsymbol{X} \frac{2}{\sqrt{\gamma}(\alpha_0 + 1)} \boldsymbol{w}_0 + \frac{\boldsymbol{z}}{2} + \boldsymbol{\epsilon} \\
&= \frac{\tilde{\alpha}_0}{2} \boldsymbol{z} + \frac{\sqrt{\gamma}}{2} \boldsymbol{X} \tilde{\boldsymbol{w}}_0 + \frac{\boldsymbol{z}}{2} + \boldsymbol{\epsilon}
\end{aligned} \tag{19}$$

Substituting $\boldsymbol{y}$ in SCLS (6) with (19):

$$\left\|\frac{\tilde{\alpha}}{2}\boldsymbol{z} + \frac{\sqrt{\gamma}}{2}\boldsymbol{X}\tilde{\boldsymbol{w}} - (\boldsymbol{y} - \frac{\boldsymbol{z}}{2})\right\|^2 = \left\|\frac{\tilde{\alpha}}{2}\boldsymbol{z} + \frac{\sqrt{\gamma}}{2}\boldsymbol{X}\tilde{\boldsymbol{w}} + \frac{\boldsymbol{z}}{2} - \left(\frac{\tilde{\alpha}_0}{2}\boldsymbol{z} + \frac{\sqrt{\gamma}}{2}\boldsymbol{X}\tilde{\boldsymbol{w}}_0 + \frac{\boldsymbol{z}}{2} + \boldsymbol{\epsilon}\right)\right\|^2$$

$$= \left\|\frac{\tilde{\alpha} - \tilde{\alpha}_0}{2}\boldsymbol{z} + \frac{\sqrt{\gamma}}{2}\boldsymbol{X}(\tilde{\boldsymbol{w}} - \tilde{\boldsymbol{w}}_0) - \boldsymbol{\epsilon}\right\|^2. \tag{20}$$

Combining formulations (6), (18), and (20), SCLS (6) is equivalent to the optimization problem:

$$\min_{\tilde{\boldsymbol{w}}, \tilde{\alpha}} \left\|\frac{\tilde{\alpha} - \tilde{\alpha}_0}{2}\boldsymbol{z} + \frac{\sqrt{\gamma}}{2}\boldsymbol{X}(\tilde{\boldsymbol{w}} - \tilde{\boldsymbol{w}}_0) - \boldsymbol{\epsilon}\right\|^2, \quad \text{s.t. } \tilde{\boldsymbol{w}}^\top \tilde{\boldsymbol{w}} + \tilde{\alpha}^2 = 1, \ \tilde{\boldsymbol{w}}_0^\top \tilde{\boldsymbol{w}}_0 + \tilde{\alpha}_0^2 = 1. \tag{21}$$

Let $\tilde{\boldsymbol{w}}^*$ denote the optimal solution to original SCLS problem (6), then, the estimation error for SCLS (21) is $\tilde{\boldsymbol{\beta}}^* := \tilde{\boldsymbol{w}}^* - \tilde{\boldsymbol{w}}_0$. To explore the optimal estimation error for SCLS (21) under different conditions of $\tilde{\alpha}_0$, we consider two scenarios: **Case 1:** If $\tilde{\alpha}_0 > 0$, set $\tilde{\alpha}(\tilde{\boldsymbol{w}}) = \sqrt{1 - \|\tilde{\boldsymbol{w}}\|^2}$. **Case 2:** If $\tilde{\alpha}_0 < 0$, set $\tilde{\alpha}(\tilde{\boldsymbol{w}}) = -\sqrt{1 - \|\tilde{\boldsymbol{w}}\|^2}$.

Both cases lead to consistent error analysis outcomes for SCLS (6), but this paper primarily discusses Case 1, with Case 2 detailed in Appendix D. According to formulation (14),

$$\tilde{\alpha}(\tilde{\boldsymbol{w}}) = \tilde{\alpha}(\tilde{\boldsymbol{w}}_0) + \frac{\partial \tilde{\alpha}(\tilde{\boldsymbol{w}})}{\partial \tilde{\boldsymbol{w}}}\Big|_{\tilde{\boldsymbol{w}} = \tilde{\boldsymbol{w}}_0} (\tilde{\boldsymbol{w}} - \tilde{\boldsymbol{w}}_0) + O(\|\tilde{\boldsymbol{w}} - \tilde{\boldsymbol{w}}_0\|^2). \tag{22}$$

The first-order approximation of $\tilde{\alpha}(\tilde{\boldsymbol{w}})$ is:

$$\hat{\tilde{\alpha}}(\tilde{\boldsymbol{w}}) = \tilde{\alpha}(\tilde{\boldsymbol{w}}_0) + \frac{\partial \tilde{\alpha}(\tilde{\boldsymbol{w}})}{\partial \tilde{\boldsymbol{w}}}\Big|_{\tilde{\boldsymbol{w}} = \tilde{\boldsymbol{w}}_0} (\tilde{\boldsymbol{w}} - \tilde{\boldsymbol{w}}_0). \tag{23}$$

When $\|\tilde{\boldsymbol{w}} - \tilde{\boldsymbol{w}}_0\| \to 0$, $\hat{\tilde{\alpha}}(\tilde{\boldsymbol{w}})$ converges to $\tilde{\alpha}(\tilde{\boldsymbol{w}})$. Using this approximation in SCLS (20) leads to an objective:

$$\left\|\frac{\hat{\tilde{\alpha}} - \tilde{\alpha}_0}{2}\boldsymbol{z} + \frac{\sqrt{\gamma}}{2}\boldsymbol{X}(\tilde{\boldsymbol{w}} - \tilde{\boldsymbol{w}}_0) - \boldsymbol{\epsilon}\right\|^2 = \left\|\frac{1}{2} \cdot \frac{\partial \tilde{\alpha}(\tilde{\boldsymbol{w}})}{\partial \tilde{\boldsymbol{w}}}\Big|_{\tilde{\boldsymbol{w}} = \tilde{\boldsymbol{w}}_0} \cdot (\tilde{\boldsymbol{w}} - \tilde{\boldsymbol{w}}_0)\boldsymbol{z} + \frac{\sqrt{\gamma}}{2}\boldsymbol{X}(\tilde{\boldsymbol{w}} - \tilde{\boldsymbol{w}}_0) - \boldsymbol{\epsilon}\right\|^2$$

$$= \frac{\sqrt{\gamma}}{2}\left\|\frac{1}{\sqrt{\gamma}} \cdot \frac{\partial \tilde{\alpha}(\tilde{\boldsymbol{w}})}{\partial \tilde{\boldsymbol{w}}}\Big|_{\tilde{\boldsymbol{w}} = \tilde{\boldsymbol{w}}_0} \cdot (\tilde{\boldsymbol{w}} - \tilde{\boldsymbol{w}}_0)\boldsymbol{z} + \boldsymbol{X}(\tilde{\boldsymbol{w}} - \tilde{\boldsymbol{w}}_0) - \frac{2\boldsymbol{\epsilon}}{\sqrt{\gamma}}\right\|^2.$$

Then, we obtain an approximated problem corresponding to SCLS (21):

$$\min_{\tilde{\boldsymbol{w}}} \frac{\sqrt{\gamma}}{2}\left\|\frac{1}{\sqrt{\gamma}} \cdot \frac{\partial \tilde{\alpha}(\tilde{\boldsymbol{w}})}{\partial \tilde{\boldsymbol{w}}}\Big|_{\tilde{\boldsymbol{w}} = \tilde{\boldsymbol{w}}_0} \cdot (\tilde{\boldsymbol{w}} - \tilde{\boldsymbol{w}}_0)\boldsymbol{z} + \boldsymbol{X}(\tilde{\boldsymbol{w}} - \tilde{\boldsymbol{w}}_0) - \frac{2\boldsymbol{\epsilon}}{\sqrt{\gamma}}\right\|^2 \tag{24}$$

Let $\hat{\tilde{\boldsymbol{w}}}^*$ denote the optimal solution to the approximated SCLS problem (24), then, the estimation error for approximated SCLS (24) is $\hat{\boldsymbol{\beta}}^* := \hat{\tilde{\boldsymbol{w}}}^* - \tilde{\boldsymbol{w}}_0$. Leveraging the simplified representation of $\hat{\tilde{\alpha}}(\tilde{\boldsymbol{w}})$, we can conduct a precise error analysis for this approximated model (24). Furthermore, if $f(\tilde{\boldsymbol{w}})$ and $\hat{f}(\tilde{\boldsymbol{w}})$ denote the objective functions of SCLS (21) and the approximated version (24), respectively,

$$f(\tilde{\boldsymbol{w}}) = \left\|\frac{\tilde{\alpha} - \tilde{\alpha}_0}{2}\boldsymbol{z} + \frac{\sqrt{\gamma}}{2}\boldsymbol{X}(\tilde{\boldsymbol{w}} - \tilde{\boldsymbol{w}}_0) - \boldsymbol{\epsilon}\right\|^2,$$

$$\hat{f}(\tilde{\boldsymbol{w}}) = \frac{\sqrt{\gamma}}{2}\left\|\frac{1}{\sqrt{\gamma}} \cdot \frac{\partial \tilde{\alpha}(\tilde{\boldsymbol{w}})}{\partial \tilde{\boldsymbol{w}}}\Big|_{\tilde{\boldsymbol{w}} = \tilde{\boldsymbol{w}}_0} \cdot (\tilde{\boldsymbol{w}} - \tilde{\boldsymbol{w}}_0)\boldsymbol{z} + \boldsymbol{X}(\tilde{\boldsymbol{w}} - \tilde{\boldsymbol{w}}_0) - \frac{2\boldsymbol{\epsilon}}{\sqrt{\gamma}}\right\|^2,$$

we have

$$\lim_{\|\tilde{\boldsymbol{w}} - \tilde{\boldsymbol{w}}_0\| \to 0} \hat{f}(\tilde{\boldsymbol{w}}) = f(\tilde{\boldsymbol{w}}). \tag{25}$$

Compared with SCLS (21), the approximation (24) is tight when $\|tildew - \tilde{\boldsymbol{w}}_0\|$ goes to 0. We later demonstrate that this convergence condition is satisfied as $n$ goes to $\infty$, independent of the original SCLS (21). This fact allows us to translate the findings about $\hat{\tilde{\boldsymbol{w}}}^*$ obtained for the approximated SCLS problem (24) to corresponding outcomes of $\tilde{\boldsymbol{w}}^*$ for the original SCLS problem (6). Given the constancy of $\gamma$, the approximated SCLS problem (24) simplifies to:

$$\min_{\tilde{\boldsymbol{\beta}}} \frac{1}{n}\left\|\boldsymbol{c}^\top \tilde{\boldsymbol{\beta}}\boldsymbol{z} + \boldsymbol{X}\tilde{\boldsymbol{\beta}} - \frac{2\boldsymbol{\epsilon}}{\sqrt{\gamma}}\right\|^2. \tag{26}$$

where $\tilde{\boldsymbol{\beta}} := \tilde{\boldsymbol{w}} - \tilde{\boldsymbol{w}}_0$, and $\boldsymbol{c} := \boldsymbol{c}(\tilde{\boldsymbol{w}}_0, \gamma) = \frac{1}{\sqrt{\gamma}} \cdot \frac{\partial \tilde{\alpha}(\tilde{\boldsymbol{w}})}{\partial \tilde{\boldsymbol{w}}}\Big|_{\tilde{\boldsymbol{w}}=\tilde{\boldsymbol{w}}_0} = \frac{1}{\sqrt{\gamma}} \cdot \frac{-\tilde{\boldsymbol{w}}_0}{\sqrt{1-\|\tilde{\boldsymbol{w}}_0\|^2}}$. The normalization of the loss function is appropriately applied, which does not alter the optimal solution. Based on the relationship (25), when $\|\tilde{\boldsymbol{\beta}}\|$ tends to 0, the approximated SCLS problem (24) effectively aligns with SCLS (21). This equivalence allows for the substitution of the analysis of the optimal cost $\tilde{\boldsymbol{\beta}}^*$ in SCLS (21) with the analysis of the optimal solution $\hat{\boldsymbol{\beta}}^*$ in the first-order optimization (26).

## 4.2 From PO to AO

A key transformation in our analysis involves converting the optimization (26) into a **PO** problem within the CGMT framework. We apply conjugate pairs (13) for optimization (26):

$$\min_{\tilde{\boldsymbol{\beta}}} \frac{1}{n} \left\| \boldsymbol{c}^\top \tilde{\boldsymbol{\beta}} \boldsymbol{z} + \boldsymbol{X}\tilde{\boldsymbol{\beta}} - \frac{2\boldsymbol{\epsilon}}{\sqrt{\gamma}} \right\|^2 = \min_{\tilde{\boldsymbol{\beta}}} \max_{\boldsymbol{u}} \frac{1}{n} \big( \boldsymbol{u}^\top \boldsymbol{X}\tilde{\boldsymbol{\beta}} + \boldsymbol{c}^\top \tilde{\boldsymbol{\beta}} \cdot \boldsymbol{u}^\top \boldsymbol{z} - \frac{2\boldsymbol{u}^\top \boldsymbol{\epsilon}}{\sqrt{\gamma}} - \frac{\|\boldsymbol{u}\|^2}{4} \big), \quad (27)$$

where $\tilde{\boldsymbol{\beta}} \in \mathbb{R}^d, \boldsymbol{u} \in \mathbb{R}^n$. Using formulations (10) and (27), the **PO** problem associated with (26) is:

$$\Phi_{\text{SCLS}}(\boldsymbol{X}) = \min_{\tilde{\boldsymbol{\beta}}} \max_{\boldsymbol{u}} \frac{1}{n} \big( \boldsymbol{u}^\top \boldsymbol{X}\tilde{\boldsymbol{\beta}} + \psi(\tilde{\boldsymbol{\beta}}, \boldsymbol{u}) \big), \quad (28)$$

where $\psi(\tilde{\boldsymbol{\beta}}, \boldsymbol{u}) := \boldsymbol{c}^\top \tilde{\boldsymbol{\beta}} \cdot \boldsymbol{u}^\top \boldsymbol{z} - \frac{2\boldsymbol{u}^\top \boldsymbol{\epsilon}}{\sqrt{\gamma}} - \frac{\|\boldsymbol{u}\|^2}{4}$. Given that the entries of $\boldsymbol{X}$ are drawn i.i.d. from $\mathcal{N}(0, 1)$ and $\psi(\tilde{\boldsymbol{\beta}}, \boldsymbol{u})$ is a convex-concave function, the **PO** problem (28) satisfies the conditions of Theorem 3.6. Consequently, we replace the challenging **PO** problem (28) with a simplified **AO** problem using CGMT:

$$\phi_{\text{SCLS}}(\boldsymbol{g}, \boldsymbol{h}) = \min_{\tilde{\boldsymbol{\beta}}} \max_{\boldsymbol{u}} \frac{1}{n} \big( \|\tilde{\boldsymbol{\beta}}\| \boldsymbol{g}^\top \boldsymbol{u} + \|\boldsymbol{u}\| \boldsymbol{h}^\top \tilde{\boldsymbol{\beta}} + \boldsymbol{c}^\top \tilde{\boldsymbol{\beta}} \cdot \boldsymbol{u}^\top \boldsymbol{z} - \frac{2\boldsymbol{u}^\top \boldsymbol{\epsilon}}{\sqrt{\gamma}} - \frac{\|\boldsymbol{u}\|^2}{4} \big)$$

$$= \min_{\tilde{\boldsymbol{\beta}}} \max_{\boldsymbol{u}} \frac{1}{n} \big[ \big( \|\tilde{\boldsymbol{\beta}}\| \boldsymbol{g} + \boldsymbol{c}^\top \tilde{\boldsymbol{\beta}} \boldsymbol{z} - \frac{2\boldsymbol{\epsilon}}{\sqrt{\gamma}} \big)^\top \boldsymbol{u} + \|\boldsymbol{u}\| \boldsymbol{h}^\top \tilde{\boldsymbol{\beta}} - \frac{\|\boldsymbol{u}\|^2}{4} \big], \quad (29)$$

where the entries of $\boldsymbol{g}$ and $\boldsymbol{h}$ are drawn i.i.d. from $\mathcal{N}(0, 1)$. Suppose $\tilde{\boldsymbol{\beta}}_{\Phi_{\text{SCLS}}}$ represents the optimal solutions of the **PO** problem (28), and $\tilde{\boldsymbol{\beta}}_{\phi_{\text{SCLS}}}$ denotes the optimal solutions of the **AO** problem (29). According to Theorem 3.6, if $\|\tilde{\boldsymbol{\beta}}_{\phi_{\text{SCLS}}}\| \xrightarrow{P} \rho^*$, then $\|\tilde{\boldsymbol{\beta}}_{\Phi_{\text{SCLS}}}\| \xrightarrow{P} \rho^*$. The reasons why **PO** (28) is more complex than **AO** (29) and the difficulties of (28) are summarized as: $(i)$ The **PO** (28) contains the matrix $\boldsymbol{X} \in \mathbb{R}^{n \times d}$ and the challenge lies in processing matrices. the **AO** (29) only contains vectors with dimensions $d$ or $n$, which are easier to handle than matrices; $(ii)$ The **AO** (29) reduces the dimension of the **PO** (28) from $n \times d$ to $\max\{d, n\}$, thereby simplifying the **PO** (28); $(iii)$ It is difficult to obtain the value that the **PO** (28) concentrates on; $(iv)$ The **AO** problem (29) can be further simplified by **AO** optimization (33) that only includes estimation error variable $\tilde{\boldsymbol{\beta}}$, which is easier to analyze than **PO** (28). These explanations enable us to effectively analyze the minimizer of the **AO** problem (29) instead of the more complex **PO** problem (28).

## 4.3 Simplification for AO

Considering that the elements of $\boldsymbol{g}$ and $\boldsymbol{z}$ are drawn i.i.d. from $\mathcal{N}(0, 1)$, and $\boldsymbol{\epsilon} \sim \mathcal{N}(0, \sigma^2 \boldsymbol{I}_d)$, the vector expression $\|\tilde{\boldsymbol{\beta}}\| \boldsymbol{g} + \boldsymbol{c}^\top \tilde{\boldsymbol{\beta}} \boldsymbol{z} - \frac{2\boldsymbol{\epsilon}}{\sqrt{\gamma}}$ in **AO** (29) behaves statistically as a random vector with entries drawn i.i.d. from $\mathcal{N}(0, \|\tilde{\boldsymbol{\beta}}\|^2 + (\boldsymbol{c}^\top \tilde{\boldsymbol{\beta}})^2 + \frac{4\sigma^2}{\gamma})$, where $\boldsymbol{I}_d$ represents a $d \times d$ identity matrix. Adopting the approach outlined by [28], we simplify the first term in **AO** (29) to $\sqrt{\|\tilde{\boldsymbol{\beta}}\|^2 + (\boldsymbol{c}^\top \tilde{\boldsymbol{\beta}})^2 + \frac{4\sigma^2}{\gamma}} \cdot \boldsymbol{g}^\top \boldsymbol{u}$:

$$\min_{\tilde{\boldsymbol{\beta}}} \max_{\boldsymbol{u}} \frac{1}{n} \Big( \sqrt{\|\tilde{\boldsymbol{\beta}}\|^2 + (\boldsymbol{c}^\top \tilde{\boldsymbol{\beta}})^2 + \frac{4\sigma^2}{\gamma}} \cdot \boldsymbol{g}^\top \boldsymbol{u} + \|\boldsymbol{u}\| \boldsymbol{h}^\top \tilde{\boldsymbol{\beta}} - \frac{\|\boldsymbol{u}\|^2}{4} \Big). \quad (30)$$

Defining $\eta = \|\boldsymbol{u}\|$ and recognizing that $\max_{\boldsymbol{u}} \boldsymbol{g}^\top \boldsymbol{u} = \|\boldsymbol{g}\| \cdot \|\boldsymbol{u}\| = \eta \|\boldsymbol{g}\|$, and considering $\boldsymbol{h} \sim \mathcal{N}(0, \boldsymbol{I}_d)$, we reformulate the optimization (30) as:

$$\min_{\tilde{\boldsymbol{\beta}}} \max_{\eta \geq 0} \frac{1}{n} \Big( \sqrt{\|\tilde{\boldsymbol{\beta}}\|^2 + (\boldsymbol{c}^\top \tilde{\boldsymbol{\beta}})^2 + \frac{4\sigma^2}{\gamma}} \cdot \|\boldsymbol{g}\| \eta + \eta \boldsymbol{h}^\top \tilde{\boldsymbol{\beta}} - \frac{\eta^2}{4} \Big). \quad (31)$$

The formulation (31) is a quadratic function of $\eta$ with the symmetric axis:

$$\eta_s = 2\Big(\sqrt{\|\tilde{\boldsymbol{\beta}}\|^2 + (\boldsymbol{c}^\top \tilde{\boldsymbol{\beta}})^2 + \frac{4\sigma^2}{\gamma}} \cdot \|\boldsymbol{g}\| + \boldsymbol{h}^\top \tilde{\boldsymbol{\beta}}\Big) > \|\tilde{\boldsymbol{\beta}}\|(\|\boldsymbol{g}\| - \|\boldsymbol{h}\|).$$

Additionally, $\eta_s(\|\boldsymbol{g}\| + \|\boldsymbol{h}\|) > \|\tilde{\boldsymbol{\beta}}\|(\|\boldsymbol{g}\|^2 - \|\boldsymbol{h}\|^2)$. Referring to [29, Lem. B.2], $\|\boldsymbol{g}\|^2$ and $\|\boldsymbol{h}\|^2$ concentrate around their means $n$ and $d$, respectively. Consequently, the value around which $\eta_s$ concentrates is nonnegative, due to $d/n < 1$. Moreover, taking $\eta_s$ into (31), the optimization objective (31) concentrates around

$$\min_{\tilde{\boldsymbol{\beta}}} \frac{1}{n} \Big(\sqrt{\|\tilde{\boldsymbol{\beta}}\|^2 + (\boldsymbol{c}^\top \tilde{\boldsymbol{\beta}})^2 + \frac{4\sigma^2}{\gamma}} \cdot \|\boldsymbol{g}\| + \boldsymbol{h}^\top \tilde{\boldsymbol{\beta}}\Big)^2$$

$$= \min_{\tilde{\boldsymbol{\beta}}} \frac{1}{n} \Big[ (\|\tilde{\boldsymbol{\beta}}\|^2 + (\boldsymbol{c}^\top \tilde{\boldsymbol{\beta}})^2 + \frac{4\sigma^2}{\gamma})\|\boldsymbol{g}\|^2 + (\boldsymbol{h}^\top \tilde{\boldsymbol{\beta}})^2 + 2\boldsymbol{h}^\top \tilde{\boldsymbol{\beta}}\|\boldsymbol{g}\| \sqrt{\|\tilde{\boldsymbol{\beta}}\|^2 + (\boldsymbol{c}^\top \tilde{\boldsymbol{\beta}})^2 + \frac{4\sigma^2}{\gamma}} \Big]. \quad (32)$$

Drawing on [29, Lem. B.2], $\|\boldsymbol{g}\|^2$, $(\boldsymbol{h}^\top \tilde{\boldsymbol{\beta}})^2$ and $\boldsymbol{h}^\top \tilde{\boldsymbol{\beta}}\|\boldsymbol{g}\|$ concentrate around their expected values: $\mathbb{E}[\|\boldsymbol{g}\|^2] = n$, $\mathbb{E}(\boldsymbol{h}^\top \tilde{\boldsymbol{\beta}})^2 = \|\tilde{\boldsymbol{\beta}}\|^2$ and $\mathbb{E}(\boldsymbol{h}^\top \tilde{\boldsymbol{\beta}}\|\boldsymbol{g}\|) = 0$. Besides, define $\Omega(\tilde{\boldsymbol{\beta}}) := \lim_{n \to \infty} \frac{\|\tilde{\boldsymbol{\beta}}\|^2}{n}$. Using analytical methods established by [27, 29, 30], as $n$ goes to $+\infty$, the optimal minimizer of (32) converges to the optimal minimizer of the following deterministic optimization in probability:

$$\min_{\tilde{\boldsymbol{\beta}}} \|\tilde{\boldsymbol{\beta}}\|^2 + (\boldsymbol{c}^\top \tilde{\boldsymbol{\beta}})^2 + \Omega(\tilde{\boldsymbol{\beta}}) + \frac{4\sigma^2}{\gamma}. \quad (33)$$

Here, we successfully reduced the complex **AO** problem (29) to a more manageable deterministic optimization problem (33), effectively focusing only on the estimation error variable $\tilde{\boldsymbol{\beta}}$ for further analysis.

### 4.4 Error Analysis

Building on the previous analysis, if the optimal solution of optimization (33) is $\|\tilde{\boldsymbol{\beta}}\| = \rho^*$, we have $\|\tilde{\boldsymbol{\beta}}_{\phi_{\mathrm{SCLS}}}\| \xrightarrow{P} \rho^*$ for **AO** problem (29). Then, by virtue of CGMT, $\|\tilde{\boldsymbol{\beta}}_{\Phi_{\mathrm{SCLS}}}\| \xrightarrow{P} \rho^*$ also holds for **PO** problem (28). If $\rho^*$ further satisfies $\rho^* = 0$, based on the relationship between the original and approximated SCLS in Section 4.1, we have $\|\tilde{\boldsymbol{w}} - \tilde{\boldsymbol{w}}_0\| \xrightarrow{P} 0$ for SCLS problems (21) and (6). Therefore, it only remains to obtain the optimal value of $\rho$ in optimization (33) that plays the role of $\|\tilde{\boldsymbol{\beta}}\|$. We conclude the estimation error analysis of the SCLS problem (6) with the following theorem.

**Theorem 4.1.** *Suppose $\tilde{\boldsymbol{w}}_0$ is the true weight parameter of the original SCLS problem (6), and $\tilde{\boldsymbol{w}}^*$ is the optimal solution to the objective function of SCLS (6). If $\lim_{n \to \infty} \frac{d}{n} \in (0, 1)$, the estimation error of SCLS (6) is given by the following probability limit:*

$$\lim_{n \to \infty} \|\tilde{\boldsymbol{w}}^* - \tilde{\boldsymbol{w}}_0\| \xrightarrow{P} 0.$$

The proof is based on the simplified **AO** problem (33) and is detailed in in Appendix B

*Remark* 4.2. Theorem 4.1 indicates that, as $n$ goes to $\infty$, the parameter vector $\tilde{\boldsymbol{w}}^*$ learned through the SCLS method (6) reliably converges to the actual parameter vector $\tilde{\boldsymbol{w}}_0$ in probability. We then can utilize Theorem 3.4 to establish the estimation error of SPG-LS (1) solved by the SCLS (6).

When applying the SCLS method (6) to solve SPG-LS (1), the validity of the solution $\boldsymbol{w}^*$ learned by the SCLS method is supported by the following theorem.

**Theorem 4.3.** *Suppose $\boldsymbol{w}_0$ is the true weight parameter of the SPG-LS (1), $\tilde{\boldsymbol{w}}^*$ is the optimal solution learned by SCLS (6), and $\boldsymbol{w}^*$ is the optimal solution recovered from $\tilde{\boldsymbol{w}}^*$ by Theorem 3.4. If $\lim_{n \to \infty} \frac{d}{n} \in (0, 1)$, the estimation error of SPG-LS (1) solved by the SCLS (6) is given by the following probability limit:*

$$\lim_{n \to \infty} \|\boldsymbol{w}^* - \boldsymbol{w}_0\| \xrightarrow{P} 0.$$

The proof relies on Theorem 4.1 and can be seen in Appendix C

*Remark* 4.4. Theorem 4.3 demonstrates that, as $n$ goes to $\infty$, the parameter vector $\boldsymbol{w}^*$ learned through the SCLS method (6) reliably converges to actual parameter vector $\boldsymbol{w}_0$ in probability. This substantiates the efficacy of the SCLS method (6) in solving SPG-LS (1).

# 5 Experiment Results

This section outlines numerical experiments conducted on synthetic datasets with high feature dimensions [31] and various levels of sparsity to validate our theoretical claims. In alignment with the methodologies described by [1, 2, 3], the true parameter $\boldsymbol{w}_0$ is generated randomly with sparsity levels set at $\frac{k}{d} = 1, 0.1, 0.01, 0.001$, where $k$ is the number of nonzero elements in $\boldsymbol{w}_0$. And, the regularization parameter $\gamma$ is set to be $0.1, 0.01$.

For each dataset $S = \{(\boldsymbol{x}_i, y_i, z_i)\}_{i=1}^n$, the input vector $\boldsymbol{x}_i$ is drawn i.i.d. from $\mathcal{N}(\boldsymbol{0}, \boldsymbol{I}_d)$, the fake output label $z_i$ is drawn i.i.d. from $\mathcal{N}(0, 1)$. Consistent with the noise model used by [2], the noise $\epsilon_i$ in our experiments is drawn i.i.d. from $\mathcal{N}(0, 0.1^2)$. According to our data generation model (2), the output labels $y_i$ are derived via:

$$\boldsymbol{x}_i^* = \arg\min_{\hat{\boldsymbol{x}}} \|\boldsymbol{w}_0^\top \hat{\boldsymbol{x}}_i - z_i\|^2 + \gamma \|\boldsymbol{x}_i - \hat{\boldsymbol{x}}_i\|^2, \quad y_i = \boldsymbol{w}_0^\top \boldsymbol{x}_i + \epsilon_i.$$

Using the samples $S = \{(\boldsymbol{x}_i, y_i, z_i)\}_{i=1}^n$, we employ the SCLS method as described by [3] to address the SPG-LS problem (1) and assess the estimation error $\|\boldsymbol{w}^* - \boldsymbol{w}_0\|$. The estimation error is averaged over 10 trials to gauge the effectiveness of the SCLS method. Results for $\frac{d}{n} = 0.5$ are shown in Figure 1. The computational resources are detailed in Appendix E.1. Additional experimental results with other parameter settings can be seen in Appendix E.2.

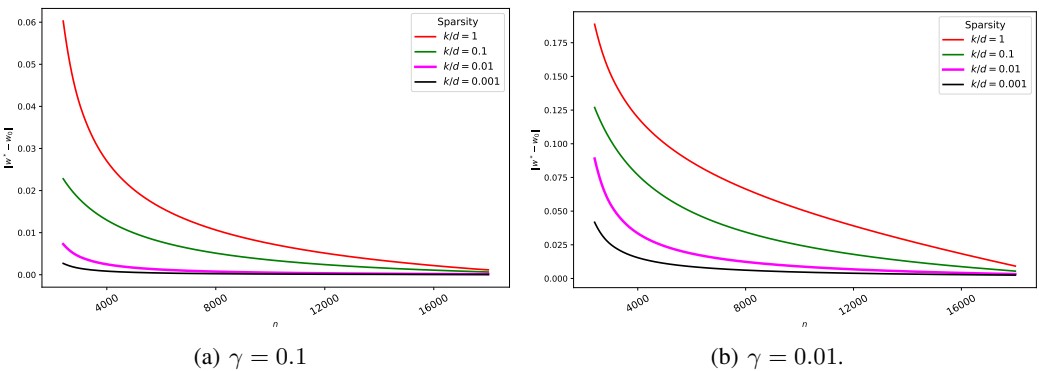

(a) $\gamma = 0.1$        (b) $\gamma = 0.01$.

Figure 1: The change of $\|\boldsymbol{w}^* - \boldsymbol{w}_0\|$ with $n$ for SCLS method under different Sparsity $k/d$.

The outcomes depicted in Figure 1 illustrate that the estimation error $\|\boldsymbol{w}^* - \boldsymbol{w}_0\|$ generated by the SCLS method decreases to 0 as $n$ goes to $\infty$. This trend corroborates the theoretical predictions established in our Theorem 4.3, thereby affirming the efficacy and reliability of the SCLS method.

# 6 Conclusion

In this paper, we apply the CGMT framework to conduct a rigorous theoretical error analysis of the SCLS method proposed by [3]. Specifically, when SCLS (6) is applied to tackle a SPG-LS model (1) with $\boldsymbol{x} \sim \mathcal{N}(0, \boldsymbol{I}_d)$, $\boldsymbol{z} \sim \mathcal{N}(0, \boldsymbol{I}_n)$ and $\epsilon \sim \mathcal{N}(0, \sigma^2)$, if $\lim_{n\to\infty} \frac{d}{n} \in (0, 1)$, we establish that:

$$\lim_{n\to\infty} \|\boldsymbol{w}^* - \boldsymbol{w}_0\| \xrightarrow{P} 0.$$

This result confirms that the learner $\boldsymbol{w}^*$ obtained through SCLS (6) accurately estimates the true learner $\boldsymbol{w}_0$ of SPG-LS model. Our empirical findings are consistent with these theoretical results. This theoretical error analysis not only validate the reliability of the SCLS method but also provides a framework for error analysis applicable to other statistical learning algorithms.

## Acknowledgments and Disclosure of Funding

This work is supported by the Key R&D Program of Hubei Province under Grant 2024BAB038, National Key R&D Program of China under Grant 2023YFC3604702, and the Fundamental Research Fund Program of LIESMARS.

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

# Error Analysis of the SCLS method in solving SPG (Appendix)

## A    Limitaion

The SCLS method is currently state-of-the-art for solving SPG-LS, having won the ICML 2022 Outstanding Paper Award [3]. However, [3] lacks theoretical analysis on the error of the SCLS method. To the best of our knowledge, we are the first to investigate the error of the SCLS method. The primary contribution of our paper is to provide a theoretical perspective on the error of the SCLS method under Gaussian assumption. It is worth emphasizing that the Gaussian hypothesis is a commonly used approach for theoretical analysis of algorithms in machine learning [7, 32]. Thus, we investigate the error of the SCLS algorithm limited to Gaussian settings. For future work, we plan to explore extensions of our findings to non-Gaussian input settings, aiming to provide insights into the universality of our results.

## B    Proof of Theorem 4.1

*Theorem* 4.1.  Suppose $\tilde{\boldsymbol{w}}_0$ is the true weight parameter of the original SCLS problem (6), and $\tilde{\boldsymbol{w}}^*$ is the optimal solution to the objective function of SCLS (6). If $\lim_{n \to \infty} \frac{d}{n} \in (0, 1)$, the estimation error of SCLS (6) is given by the following probability limit:

$$\lim_{n \to \infty} \|\tilde{\boldsymbol{w}}^* - \tilde{\boldsymbol{w}}_0\| \xrightarrow{P} 0.$$

*Proof.*  Given the simplified **AO** problem (33), define

$$\Gamma(\tilde{\boldsymbol{\beta}}) := \|\tilde{\boldsymbol{\beta}}\|^2 + (\boldsymbol{c}^\top \tilde{\boldsymbol{\beta}})^2 + \frac{4\sigma^2}{\gamma} + \Omega(\tilde{\boldsymbol{\beta}}). \tag{34}$$

Let $\hat{\boldsymbol{\beta}}^*$ be the optimal solution to $\Gamma(\tilde{\boldsymbol{\beta}})$, then

$$\frac{\partial \Gamma(\tilde{\boldsymbol{\beta}})}{\partial \tilde{\boldsymbol{\beta}}} = 2\tilde{\boldsymbol{\beta}} + 2(\boldsymbol{c}^\top \tilde{\boldsymbol{\beta}})\boldsymbol{c} + \lim_{n \to \infty} \frac{2\tilde{\boldsymbol{\beta}}}{n} = \boldsymbol{0} \Rightarrow \hat{\boldsymbol{\beta}}^* = \boldsymbol{0}. \tag{35}$$

If $\lambda$ represents the eigenvalue of $\boldsymbol{cc}^\top$,

$$|\lambda \boldsymbol{I}_d - \boldsymbol{cc}^\top| = \lambda^{d-1}|\lambda - \boldsymbol{c}^\top \boldsymbol{c}| = 0 \Rightarrow \lambda = 0 \text{ or } \|\boldsymbol{c}\|^2.$$

Then, the Hessian matrix $\nabla^2 \Gamma(\tilde{\boldsymbol{\beta}}) = \frac{\partial^2 \Gamma}{\partial \tilde{\boldsymbol{\beta}}^2}$ satisfies:

$$\frac{1}{2}\nabla^2 \Gamma(\tilde{\boldsymbol{\beta}}) = \boldsymbol{I}_d + \begin{pmatrix} c_1^2 & c_1 c_2 & \cdots & c_1 c_d \\ c_2 c_1 & c_2^2 & \cdots & c_2 c_d \\ \vdots & \vdots & \ddots & \vdots \\ c_d c_1 & c_d c_2 & \cdots & c_d^2 \end{pmatrix} + \lim_{n \to \infty} \frac{1}{n}\boldsymbol{I}_d = (1 + \lim_{n \to \infty} \frac{1}{n})\boldsymbol{I}_d + \boldsymbol{cc}^\top \succ \boldsymbol{0}. \tag{36}$$

This indicates that $\nabla^2 \Gamma(\tilde{\boldsymbol{\beta}})$ is positive definite matrix. Consequently, according to [33], $\Gamma(\tilde{\boldsymbol{\beta}})$ is a strongly convex function of $\tilde{\boldsymbol{\beta}}$, and $\hat{\boldsymbol{\beta}}^* = \boldsymbol{0}$ is the unique global minimum. Moreover, we note:

$$\|\hat{\boldsymbol{\beta}}^*\| = 0 \quad \text{and} \quad \Gamma(\hat{\boldsymbol{\beta}}^*) = \frac{4\sigma^2}{\gamma}. \tag{37}$$

Combining (36) and (37), the **AO** problem (33) satisfies the conditions of CGMT. Since $\|\hat{\boldsymbol{\beta}}^*\| \to 0$ occurs when $n \to \infty$, the analysis of (25) holds:

$$\lim_{n \to \infty} \hat{f}(\tilde{\boldsymbol{w}}) = f(\tilde{\boldsymbol{w}}). \tag{38}$$

Formulation (38) allows us translate the analysis on the optimal solution $\hat{\tilde{\boldsymbol{w}}}^*$ of the approximated SCLS problem (24) to the analysis on corresponding optimal solution $\tilde{\boldsymbol{w}}^*$ of the original SCLS

problem (21). Therefore, based on the relationship of SCLS problems (21) and (6), by applying the principles of CGMT, we conclude:

$$\lim_{n \to \infty} \|\tilde{\boldsymbol{\beta}}^*\| \xrightarrow{P} 0 \Leftrightarrow \lim_{n \to \infty} \|\tilde{\boldsymbol{w}}^* - \tilde{\boldsymbol{w}}_0\| \xrightarrow{P} 0.$$

$\square$

## C   Proof of Theorem 4.3

*Theorem* 4.3. Suppose $\boldsymbol{w}_0$ is the true weight parameter of the SPG-LS (1), $\tilde{\boldsymbol{w}}^*$ is the optimal solution learned by SCLS (6), and $\boldsymbol{w}^*$ is the optimal solution recovered from $\tilde{\boldsymbol{w}}^*$ by Theorem 3.4. If $\lim_{n \to \infty} \frac{d}{n} \in (0, 1)$, the estimation error of SPG-LS (1) solved by the SCLS (6) is given by the following probability limit:

$$\lim_{n \to \infty} \|\boldsymbol{w}^* - \boldsymbol{w}_0\| \xrightarrow{P} 0.$$

*Proof.* According to Theorem 3.4 and formulation (17),

$$\boldsymbol{w} = \frac{\sqrt{\gamma}}{1 - \tilde{\alpha}} \tilde{\boldsymbol{w}}, \quad \boldsymbol{w}_0 = \frac{\sqrt{\gamma}}{1 - \tilde{\alpha}_0} \tilde{\boldsymbol{w}}_0.$$

Then, the difference between the estimated and true parameter vectors can be expressed as:

$$\|\boldsymbol{w}^* - \boldsymbol{w}_0\| = \left\| \frac{\sqrt{\gamma}}{1 - \tilde{\alpha}^*} \tilde{\boldsymbol{w}}^* - \frac{\sqrt{\gamma}}{1 - \tilde{\alpha}_0} \tilde{\boldsymbol{w}}_0 \right\|. \tag{39}$$

Define

$$\tau(\tilde{\boldsymbol{w}}) := \frac{\sqrt{\gamma}}{1 - \tilde{\alpha}} \tilde{\boldsymbol{w}} = \frac{\sqrt{\gamma}}{1 - \sqrt{1 - \|\tilde{\boldsymbol{w}}\|^2}} \tilde{\boldsymbol{w}}. \tag{40}$$

Based on (14),

$$\tau(\tilde{\boldsymbol{w}}) = \tau(\tilde{\boldsymbol{w}}_0) + \frac{\partial \tau(\tilde{\boldsymbol{w}})}{\partial \tilde{\boldsymbol{w}}} \bigg|_{\tilde{\boldsymbol{w}} = \tilde{\boldsymbol{w}}_0} (\tilde{\boldsymbol{w}} - \tilde{\boldsymbol{w}}_0) + O(\|\tilde{\boldsymbol{w}} - \tilde{\boldsymbol{w}}_0\|^2). \tag{41}$$

Combining (39) $\sim$ (41), we get:

$$\begin{aligned} \|\boldsymbol{w}^* - \boldsymbol{w}_0\| =& \left\| \frac{\partial \tau(\tilde{\boldsymbol{w}})}{\partial \tilde{\boldsymbol{w}}} \bigg|_{\tilde{\boldsymbol{w}} = \tilde{\boldsymbol{w}}_0} (\tilde{\boldsymbol{w}}^* - \tilde{\boldsymbol{w}}_0) + O(\|\tilde{\boldsymbol{w}}^* - \tilde{\boldsymbol{w}}_0\|^2) \right\| \\ \leq& \left\| \frac{\partial \tau(\tilde{\boldsymbol{w}})}{\partial \tilde{\boldsymbol{w}}} \bigg|_{\tilde{\boldsymbol{w}} = \tilde{\boldsymbol{w}}_0} \right\| \cdot \|\tilde{\boldsymbol{w}}^* - \tilde{\boldsymbol{w}}_0\| + O(\|\tilde{\boldsymbol{w}} - \tilde{\boldsymbol{w}}_0\|^2). \end{aligned}$$

Invoking Theorem 4.1, we know:

$$\lim_{n \to \infty} \|\tilde{\boldsymbol{w}}^* - \tilde{\boldsymbol{w}}_0\| \xrightarrow{P} 0,$$

which implies:

$$\lim_{n \to \infty} \|\boldsymbol{w}^* - \boldsymbol{w}_0\| \xrightarrow{P} 0.$$

$\square$

## D   The Error Analysis in Case 2

**Case 2:** If $\tilde{\alpha}_0 < 0$, we set $\tilde{\alpha}(\tilde{\boldsymbol{w}}) = -\sqrt{1 - \|\tilde{\boldsymbol{w}}\|^2}$. Consequently, $\tilde{\alpha}_0(\tilde{\boldsymbol{w}}) = -\sqrt{1 - \|\tilde{\boldsymbol{w}}_0\|^2}$. It is important to note that although the expressions for $\tilde{\alpha}(\tilde{\boldsymbol{w}})$ and $\tilde{\alpha}_0(\tilde{\boldsymbol{w}})$ differ from those in **Case 1**, the approach and methodology for the error analysis of the SCLS (6) in **Case 2** remain analogous to those outlined in **Case 1**.

## D.1 From the SCLS Method to PO

According to formulation (14),

$$\tilde{\alpha}(\tilde{\boldsymbol{w}}) = \tilde{\alpha}(\tilde{\boldsymbol{w}}_0) + \frac{\partial \tilde{\alpha}(\tilde{\boldsymbol{w}})}{\partial \tilde{\boldsymbol{w}}}\Big|_{\tilde{\boldsymbol{w}}=\tilde{\boldsymbol{w}}_0}(\tilde{\boldsymbol{w}} - \tilde{\boldsymbol{w}}_0) + O(\|\tilde{\boldsymbol{w}} - \tilde{\boldsymbol{w}}_0\|^2). \tag{42}$$

The first-order approximation of $\tilde{\alpha}(\tilde{\boldsymbol{w}})$ is

$$\hat{\tilde{\alpha}}(\tilde{\boldsymbol{w}}) = \tilde{\alpha}(\tilde{\boldsymbol{w}}_0) + \frac{\partial \tilde{\alpha}(\tilde{\boldsymbol{w}})}{\partial \tilde{\boldsymbol{w}}}\Big|_{\tilde{\boldsymbol{w}}=\tilde{\boldsymbol{w}}_0}(\tilde{\boldsymbol{w}} - \tilde{\boldsymbol{w}}_0). \tag{43}$$

If $\|\tilde{\boldsymbol{w}} - \tilde{\boldsymbol{w}}_0\| \to 0$, we have $\hat{\tilde{\alpha}}(\tilde{\boldsymbol{w}}) = \tilde{\alpha}(\tilde{\boldsymbol{w}})$. Substituting $\tilde{\alpha}(\tilde{\boldsymbol{w}})$ in (20) by its first-order approximation (23),

$$\begin{aligned}
&\left\|\frac{\hat{\tilde{\alpha}} - \tilde{\alpha}_0}{2}\boldsymbol{z} + \frac{\sqrt{\gamma}}{2}\boldsymbol{X}(\tilde{\boldsymbol{w}} - \tilde{\boldsymbol{w}}_0) - \boldsymbol{\epsilon}\right\|^2 \\
={}&\left\|\frac{1}{2} \cdot \frac{\partial \tilde{\alpha}(\tilde{\boldsymbol{w}})}{\partial \tilde{\boldsymbol{w}}}\Big|_{\tilde{\boldsymbol{w}}=\tilde{\boldsymbol{w}}_0} \cdot (\tilde{\boldsymbol{w}} - \tilde{\boldsymbol{w}}_0)\boldsymbol{z} + \frac{\sqrt{\gamma}}{2}\boldsymbol{X}(\tilde{\boldsymbol{w}} - \tilde{\boldsymbol{w}}_0) - \boldsymbol{\epsilon}\right\|^2 \\
={}&\frac{\sqrt{\gamma}}{2}\left\|\frac{1}{\sqrt{\gamma}} \cdot \frac{\partial \tilde{\alpha}(\tilde{\boldsymbol{w}})}{\partial \tilde{\boldsymbol{w}}}\Big|_{\tilde{\boldsymbol{w}}=\tilde{\boldsymbol{w}}_0} \cdot (\tilde{\boldsymbol{w}} - \tilde{\boldsymbol{w}}_0)\boldsymbol{z} + \boldsymbol{X}(\tilde{\boldsymbol{w}} - \tilde{\boldsymbol{w}}_0) - \frac{2\boldsymbol{\epsilon}}{\sqrt{\gamma}}\right\|^2.
\end{aligned}$$

Then, we obtain an approximated SCLS problem corresponding to SCLS (21):

$$\min_{\tilde{\boldsymbol{w}}} \frac{\sqrt{\gamma}}{2}\left\|\frac{1}{\sqrt{\gamma}} \cdot \frac{\partial \tilde{\alpha}(\tilde{\boldsymbol{w}})}{\partial \tilde{\boldsymbol{w}}}\Big|_{\tilde{\boldsymbol{w}}=\tilde{\boldsymbol{w}}_0} \cdot (\tilde{\boldsymbol{w}} - \tilde{\boldsymbol{w}}_0)\boldsymbol{z} + \boldsymbol{X}(\tilde{\boldsymbol{w}} - \tilde{\boldsymbol{w}}_0) - \frac{2\boldsymbol{\epsilon}}{\sqrt{\gamma}}\right\|^2 \tag{44}$$

Denote $\hat{\tilde{\boldsymbol{w}}}^*$ as the optimal solution to the approximated SCLS problem (44). Then, the estimation error for approximated SCLS (44) is $\hat{\boldsymbol{\beta}}^* := \hat{\tilde{\boldsymbol{w}}}^* - \tilde{\boldsymbol{w}}_0$. Taking advantage of the simple characterization of $\hat{\tilde{\alpha}}(\tilde{\boldsymbol{w}})$, we can accurately analyze the error in the resulting approximated SCLS problem (44). In other words, if $f(\tilde{\boldsymbol{w}})$ and $\hat{f}(\tilde{\boldsymbol{w}})$ denote the objective functions of SCLS (21) and the approximated SCLS (44), respectively,

$$\begin{aligned}
f(\tilde{\boldsymbol{w}}) ={}&\left\|\frac{\tilde{\alpha} - \tilde{\alpha}_0}{2}\boldsymbol{z} + \frac{\sqrt{\gamma}}{2}\boldsymbol{X}(\tilde{\boldsymbol{w}} - \tilde{\boldsymbol{w}}_0) - \boldsymbol{\epsilon}\right\|^2, \\
\hat{f}(\tilde{\boldsymbol{w}}) ={}&\frac{\sqrt{\gamma}}{2}\left\|\frac{1}{\sqrt{\gamma}} \cdot \frac{\partial \tilde{\alpha}(\tilde{\boldsymbol{w}})}{\partial \tilde{\boldsymbol{w}}}\Big|_{\tilde{\boldsymbol{w}}=\tilde{\boldsymbol{w}}_0} \cdot (\tilde{\boldsymbol{w}} - \tilde{\boldsymbol{w}}_0)\boldsymbol{z} + \boldsymbol{X}(\tilde{\boldsymbol{w}} - \tilde{\boldsymbol{w}}_0) - \frac{2\boldsymbol{\epsilon}}{\sqrt{\gamma}}\right\|^2,
\end{aligned}$$

we have

$$\lim_{\|\tilde{\boldsymbol{w}}-\tilde{\boldsymbol{w}}_0\|\to 0} \hat{f}(\tilde{\boldsymbol{w}}) = f(\tilde{\boldsymbol{w}}). \tag{45}$$

Compared with SCLS (21), the approximation (44) is tight when $\|\tilde{\boldsymbol{w}} - \tilde{\boldsymbol{w}}_0\| \to 0$, and we later demonstrate that this case occurs independent on the original SCLS (21) as $n \to +\infty$. This fact allows us to translate the findings about $\hat{\tilde{\boldsymbol{w}}}^*$ obtained for the approximated SCLS problem (44) to corresponding precise outcomes of $\tilde{\boldsymbol{w}}^*$ for the original SCLS problem (6).

Because $\gamma$ is a constant, the approximated SCLS problem (44) is equivalent to the following optimization problem:

$$\min_{\tilde{\boldsymbol{\beta}}} \frac{1}{n}\left\|\boldsymbol{c}^\top \tilde{\boldsymbol{\beta}}\boldsymbol{z} + \boldsymbol{X}\tilde{\boldsymbol{\beta}} - \frac{2\boldsymbol{\epsilon}}{\sqrt{\gamma}}\right\|^2. \tag{46}$$

where the optimization variable is changed from $\tilde{\boldsymbol{w}}$ to $\tilde{\boldsymbol{\beta}} := \tilde{\boldsymbol{w}} - \tilde{\boldsymbol{w}}_0$, and $\boldsymbol{c} := \boldsymbol{c}(\tilde{\boldsymbol{w}}_0, \gamma) = \frac{1}{\sqrt{\gamma}} \cdot \frac{\partial \tilde{\alpha}(\tilde{\boldsymbol{w}})}{\partial \tilde{\boldsymbol{w}}}\Big|_{\tilde{\boldsymbol{w}}=\tilde{\boldsymbol{w}}_0} = \frac{1}{\sqrt{\gamma}} \cdot \frac{\tilde{\boldsymbol{w}}_0}{\sqrt{1-\|\tilde{\boldsymbol{w}}_0\|^2}}$. The normalization of the loss function is appropriately applied, which does not alter the optimal solution. Based on the analysis on formulation (45), when $\|\tilde{\boldsymbol{\beta}}\| \to 0$, the approximated SCLS problem (44) becomes equivalent to SCLS (21). Consequently, the analysis of the optimal cost $\tilde{\boldsymbol{\beta}}^*$ in SCLS (21) can be replaced by the analysis of the optimal solution $\hat{\boldsymbol{\beta}}^*$ in optimization (46).

## D.2 From PO to AO

The crucial step involves transforming optimization (46) into a **PO** problem within the CGMT framework. We utilize conjugate pairs (13) for optimization (46):

$$\min_{\tilde{\beta}} \frac{1}{n} \left\| c^\top \tilde{\beta} z + X\tilde{\beta} - \frac{2\epsilon}{\sqrt{\gamma}} \right\|^2 = \min_{\tilde{\beta}} \max_{u} \frac{1}{n} \left( u^\top X\tilde{\beta} + c^\top \tilde{\beta} \cdot u^\top z - \frac{2u^\top \epsilon}{\sqrt{\gamma}} - \frac{\|u\|^2}{4} \right), \quad (47)$$

where $\tilde{\beta} \in \mathbb{R}^d, u \in \mathbb{R}^n$. Using formulations (10) and (47), the **PO** problem associated with the estimation error of the approximate SCLS (46) is

$$\Phi_{\text{SCLS}}(X) = \min_{\tilde{\beta}} \max_{u} \frac{1}{n} \left( u^\top X\tilde{\beta} + \psi(\tilde{\beta}, u) \right), \quad (48)$$

where

$$\psi(\tilde{\beta}, u) := c^\top \tilde{\beta} \cdot u^\top z - \frac{2u^\top \epsilon}{\sqrt{\gamma}} - \frac{\|u\|^2}{4}.$$

Given that the entries of $X$ are drawn i.i.d. from $\mathcal{N}(0,1)$ and $\psi(\tilde{\beta}, u)$ is a convex-concave function, the **PO** problem (48) meets the assumptions of Theorem 3.6. Consequently, we replace the challenging **PO** problem (48) with a simplified **AO** problem using CGMT:

$$\phi_{\text{SCLS}}(g, h) = \min_{\tilde{\beta}} \max_{u} \frac{1}{n} \left( \|\tilde{\beta}\| g^\top u + \|u\| h^\top \tilde{\beta} + c^\top \tilde{\beta} \cdot u^\top z - \frac{2u^\top \epsilon}{\sqrt{\gamma}} - \frac{\|u\|^2}{4} \right)$$

$$= \min_{\tilde{\beta}} \max_{u} \frac{1}{n} \left[ \left( \|\tilde{\beta}\| g + c^\top \tilde{\beta} z - \frac{2\epsilon}{\sqrt{\gamma}} \right)^\top u + \|u\| h^\top \tilde{\beta} - \frac{\|u\|^2}{4} \right], \quad (49)$$

where the entries of $g$, $h$ are drawn i.i.d. from $\mathcal{N}(0,1)$. Suppose $\tilde{\beta}_{\Phi_{\text{SCLS}}}$ represents the optimal solutions of the **PO** problem (48), and $\tilde{\beta}_{\phi_{\text{SCLS}}}$ denotes the optimal solutions of the **AO** problem (49). According to Theorem 3.6, if $\|\tilde{\beta}_{\phi_{\text{SCLS}}}\| \xrightarrow{P} \rho^*$, we have $\|\tilde{\beta}_{\Phi_{\text{SCLS}}}\| \xrightarrow{P} \rho^*$. Thus, we can analyze the minimizer of the **AO** problem (49) instead of the **PO** problem (48).

## D.3 Simplification for AO

Given that the entries of $g$ and $z$ are drawn i.i.d. from $\mathcal{N}(0,1)$, and $\epsilon \sim \mathcal{N}(0, \sigma^2 I_d)$, the expression $\|\tilde{\beta}\| g + c^\top \tilde{\beta} z - \frac{2\epsilon}{\sqrt{\gamma}}$ in **AO** (49) is statistically identical to a random vector with entries drawn i.i.d. from $\mathcal{N}(0, \|\tilde{\beta}\|^2 + (c^\top \tilde{\beta})^2 + \frac{4\sigma^2}{\gamma})$, where $I_d$ represents a $d \times d$ identity matrix. Following the methodology outlined by [28], we substitute the first term in **AO** (49) with $\sqrt{\|\tilde{\beta}\|^2 + (c^\top \tilde{\beta})^2 + \frac{4\sigma^2}{\gamma}} \cdot g^\top u$. This yields:

$$\min_{\tilde{\beta}} \max_{u} \frac{1}{n} \left( \sqrt{\|\tilde{\beta}\|^2 + (c^\top \tilde{\beta})^2 + \frac{4\sigma^2}{\gamma}} \cdot g^\top u + \|u\| h^\top \tilde{\beta} - \frac{\|u\|^2}{4} \right). \quad (50)$$

Let $\eta = \|u\|$. Since $\max_u g^\top u = \|g\| \cdot \|u\| = \eta \|g\|$, and $h \sim \mathcal{N}(0, I_d)$, optimization (50) can be equivalently expressed as:

$$\min_{\tilde{\beta}} \max_{\eta \geq 0} \frac{1}{n} \left( \sqrt{\|\tilde{\beta}\|^2 + (c^\top \tilde{\beta})^2 + \frac{4\sigma^2}{\gamma}} \cdot \|g\| \eta + \eta h^\top \tilde{\beta} - \frac{\eta^2}{4} \right). \quad (51)$$

The formulation (51) is a quadratic function of $\eta$ with the symmetric axis:

$$\eta_s = 2 \left( \sqrt{\|\tilde{\beta}\|^2 + (c^\top \tilde{\beta})^2 + \frac{4\sigma^2}{\gamma}} \cdot \|g\| + h^\top \tilde{\beta} \right) > \|\tilde{\beta}\| (\|g\| - \|h\|).$$

Furthermore, $\eta_s(\|g\| + \|h\|) > \|\tilde{\beta}\|(\|g\|^2 - \|h\|^2)$. According to [29, Lem. B.2], $\|g\|^2$ and $\|h\|^2$ concentrate around their means $n$ and $d$, respectively. Thus, the value that $\eta_s$ concentrates around is

nonnegative. Additionally, taking $\eta_s$ into (31), the objective (51) concentrates around

$$\min_{\tilde{\boldsymbol{\beta}}} \frac{1}{n}\left(\sqrt{\|\tilde{\boldsymbol{\beta}}\|^2 + (\boldsymbol{c}^\top\tilde{\boldsymbol{\beta}})^2 + \frac{4\sigma^2}{\gamma}} \cdot \|\boldsymbol{g}\| + \boldsymbol{h}^\top\tilde{\boldsymbol{\beta}}\right)^2$$

$$= \min_{\tilde{\boldsymbol{\beta}}} \frac{1}{n}\left[(\|\tilde{\boldsymbol{\beta}}\|^2 + (\boldsymbol{c}^\top\tilde{\boldsymbol{\beta}})^2 + \frac{4\sigma^2}{\gamma})\|\boldsymbol{g}\|^2 + (\boldsymbol{h}^\top\tilde{\boldsymbol{\beta}})^2 + 2\boldsymbol{h}^\top\tilde{\boldsymbol{\beta}}\|\boldsymbol{g}\|\sqrt{\|\tilde{\boldsymbol{\beta}}\|^2 + (\boldsymbol{c}^\top\tilde{\boldsymbol{\beta}})^2 + \frac{4\sigma^2}{\gamma}}\right], \quad (52)$$

According to [29, Lem. B.2], $\|\boldsymbol{g}\|^2$, $(\boldsymbol{h}^\top\tilde{\boldsymbol{\beta}})^2$ and $\boldsymbol{h}^\top\tilde{\boldsymbol{\beta}}\|\boldsymbol{g}\|$ concentrate around their means: $\mathbb{E}[\|\boldsymbol{g}\|^2] = n$, $\mathbb{E}(\boldsymbol{h}^\top\tilde{\boldsymbol{\beta}})^2 = \|\tilde{\boldsymbol{\beta}}\|^2$ and $\mathbb{E}(\boldsymbol{h}^\top\tilde{\boldsymbol{\beta}}\|\boldsymbol{g}\|) = 0$. Besides, define $\Omega(\tilde{\boldsymbol{\beta}}) := \lim_{n\to\infty} \frac{\|\tilde{\boldsymbol{\beta}}\|^2}{n}$. Following the methodology used by [27, 29, 30], as $n \to +\infty$, the optimal minimizer of (52) converges to the optimal minimizer of the following deterministic optimization in probability:

$$\min_{\tilde{\boldsymbol{\beta}}} \|\tilde{\boldsymbol{\beta}}\|^2 + (\boldsymbol{c}^\top\tilde{\boldsymbol{\beta}})^2 + \Omega(\tilde{\boldsymbol{\beta}}) + \frac{4\sigma^2}{\gamma}. \quad (53)$$

Here, we complete the simplifications by reducing the **AO** problem (49) to an equivalent optimization (53) that now only involves the estimation error variable $\tilde{\boldsymbol{\beta}}$.

### D.4  Error Analysis

Building on the previous analysis, if the optimal solution of optimization (53) is $\|\tilde{\boldsymbol{\beta}}\| = \rho^*$, we have $\|\tilde{\boldsymbol{\beta}}_{\phi_{\mathrm{SCLS}}}\| \xrightarrow{P} \rho^*$ for **AO** problem (49). Then, by virtue of CGMT, $\|\tilde{\boldsymbol{\beta}}_{\Phi_{\mathrm{SCLS}}}\| \xrightarrow{P} \rho^*$ also holds for **PO** problem (48). If $\rho^*$ further satisfies $\rho^* = 0$, based on the relationship between the original and approximated SCLS in Section 4.1, we have $\|\tilde{\boldsymbol{w}} - \tilde{\boldsymbol{w}}_0\| \xrightarrow{P} 0$ for SCLS problems (21) and (6). Therefore, it only remains to obtain the optimal value of $\rho$ in optimization (53) that plays the role of $\|\tilde{\boldsymbol{\beta}}\|$. We conclude the estimation error analysis of the SCLS problem (6) with the following theorem.

**Theorem D.1.** *Suppose $\tilde{\boldsymbol{w}}_0$ is the true weight parameter of the original SCLS problem (6), and $\tilde{\boldsymbol{w}}^*$ is the optimal solution to the objective function of SCLS (6). If $\lim_{n\to\infty} \frac{d}{n} \in (0, 1)$, the estimation error of SCLS (6) is given by the following probability limit:*

$$\lim_{n\to\infty} \|\tilde{\boldsymbol{w}}^* - \tilde{\boldsymbol{w}}_0\| \xrightarrow{P} 0.$$

*Proof.* Given the simplified **AO** problem (53), define

$$\Gamma(\tilde{\boldsymbol{\beta}}) := \|\tilde{\boldsymbol{\beta}}\|^2 + (\boldsymbol{c}^\top\tilde{\boldsymbol{\beta}})^2 + \frac{4\sigma^2}{\gamma} + \Omega(\tilde{\boldsymbol{\beta}}). \quad (54)$$

Let $\hat{\boldsymbol{\beta}}^*$ be the optimal solution to $\Gamma(\tilde{\boldsymbol{\beta}})$, then

$$\frac{\partial \Gamma(\tilde{\boldsymbol{\beta}})}{\partial \tilde{\boldsymbol{\beta}}} = 2\tilde{\boldsymbol{\beta}} + 2(\boldsymbol{c}^\top\tilde{\boldsymbol{\beta}})\boldsymbol{c} + \lim_{n\to\infty} \frac{2\tilde{\boldsymbol{\beta}}}{n} = \mathbf{0} \Rightarrow \hat{\boldsymbol{\beta}}^* = \mathbf{0}. \quad (55)$$

If $\lambda$ represents the eigenvalue of $\boldsymbol{c}\boldsymbol{c}^\top$,

$$|\lambda\boldsymbol{I}_d - \boldsymbol{c}\boldsymbol{c}^\top| = \lambda^{d-1}|\lambda - \boldsymbol{c}^\top\boldsymbol{c}| = 0 \Rightarrow \lambda = 0 \text{ or } \|\boldsymbol{c}\|^2.$$

Then, the Hessian matrix $\nabla^2\Gamma(\tilde{\boldsymbol{\beta}}) = \frac{\partial^2\Gamma}{\partial\tilde{\boldsymbol{\beta}}^2}$ satisfies:

$$\frac{1}{2}\nabla^2\Gamma(\tilde{\boldsymbol{\beta}}) = (1 + \lim_{n\to\infty} \frac{1}{n})\boldsymbol{I}_d + \boldsymbol{c}\boldsymbol{c}^\top \succ \mathbf{0}. \quad (56)$$

This indicates that $\nabla^2\Gamma(\tilde{\boldsymbol{\beta}})$ is positive definite matrix. Consequently, according to [33], $\Gamma(\tilde{\boldsymbol{\beta}})$ is a strongly convex function of $\tilde{\boldsymbol{\beta}}$, and $\hat{\boldsymbol{\beta}}^* = \mathbf{0}$ is the unique global minimum. Moreover, we note:

$$\|\hat{\boldsymbol{\beta}}^*\| = 0 \quad \text{and} \quad \Gamma(\hat{\boldsymbol{\beta}}^*) = \frac{4\sigma^2}{\gamma}. \quad (57)$$

Combining (56) and (57), the **AO** problem (53) satisfies the conditions of CGMT. Since $\|\hat{\tilde{\boldsymbol{\beta}}}^*\| \to 0$ occurs when $n$ goes to $\infty$, the analysis of (25) holds:

$$\lim_{n \to \infty} \hat{f}(\tilde{\boldsymbol{w}}) = f(\tilde{\boldsymbol{w}}). \tag{58}$$

Formulation (58) allows us translate the analysis on the optimal solution $\hat{\tilde{\boldsymbol{w}}}^*$ of the approximated SCLS problem (44) to the analysis on corresponding optimal solution $\tilde{\boldsymbol{w}}^*$ of the original SCLS problem (21). Therefore, by applying the principles of CGMT, we conclude:

$$\lim_{n \to \infty} \|\tilde{\boldsymbol{\beta}}^*\| \xrightarrow{P} 0 \Leftrightarrow \lim_{n \to \infty} \|\tilde{\boldsymbol{w}}^* - \tilde{\boldsymbol{w}}_0\| \xrightarrow{P} 0.$$

$\square$

When applying the SCLS method (6) to solve SPG-LS (1), the reliability of the SCLS method (6) in solving SPG-LS (1) is supported by the following theorem..

**Theorem D.2.** *Suppose $\boldsymbol{w}_0$ is the true weight parameter of the SPG-LS (1), $\tilde{\boldsymbol{w}}^*$ is the optimal solution learned by SCLS (6), and $\boldsymbol{w}^*$ is the optimal solution recovered from $\tilde{\boldsymbol{w}}^*$ by Theorem 3.4. If $\lim_{n \to \infty} \frac{d}{n} \in (0, 1)$, the estimation error of SPG-LS (1) solved by the SCLS (6) is given by the following probability limit:*

$$\lim_{n \to \infty} \|\boldsymbol{w}^* - \boldsymbol{w}_0\| \xrightarrow{P} 0.$$

*Proof.* According to Theorem 3.4 and formulation (17),

$$\boldsymbol{w} = \frac{\sqrt{\gamma}}{1 - \tilde{\alpha}} \tilde{\boldsymbol{w}}, \quad \boldsymbol{w}_0 = \frac{\sqrt{\gamma}}{1 - \tilde{\alpha}_0} \tilde{\boldsymbol{w}}_0.$$

Then, the difference between the estimated and true parameter vectors can be expressed as:

$$\|\boldsymbol{w}^* - \boldsymbol{w}_0\| = \left\| \frac{\sqrt{\gamma}}{1 - \tilde{\alpha}^*} \tilde{\boldsymbol{w}}^* - \frac{\sqrt{\gamma}}{1 - \tilde{\alpha}_0} \tilde{\boldsymbol{w}}_0 \right\|. \tag{59}$$

Define

$$\tau(\tilde{\boldsymbol{w}}) := \frac{\sqrt{\gamma}}{1 - \tilde{\alpha}} \tilde{\boldsymbol{w}} = \frac{\sqrt{\gamma}}{1 + \sqrt{1 - \|\tilde{\boldsymbol{w}}\|^2}} \tilde{\boldsymbol{w}}. \tag{60}$$

Based on (14),

$$\tau(\tilde{\boldsymbol{w}}) = \tau(\tilde{\boldsymbol{w}}_0) + \frac{\partial \tau(\tilde{\boldsymbol{w}})}{\partial \tilde{\boldsymbol{w}}} \bigg|_{\tilde{\boldsymbol{w}} = \tilde{\boldsymbol{w}}_0} (\tilde{\boldsymbol{w}} - \tilde{\boldsymbol{w}}_0) + O(\|\tilde{\boldsymbol{w}} - \tilde{\boldsymbol{w}}_0\|^2). \tag{61}$$

Combining (59) $\sim$ (61), we get:

$$\begin{aligned} \|\boldsymbol{w}^* - \boldsymbol{w}_0\| &= \left\| \frac{\partial \tau(\tilde{\boldsymbol{w}})}{\partial \tilde{\boldsymbol{w}}} \bigg|_{\tilde{\boldsymbol{w}} = \tilde{\boldsymbol{w}}_0} (\tilde{\boldsymbol{w}}^* - \tilde{\boldsymbol{w}}_0) + O(\|\tilde{\boldsymbol{w}}^* - \tilde{\boldsymbol{w}}_0\|^2) \right\| \\ &\leq \left\| \frac{\partial \tau(\tilde{\boldsymbol{w}})}{\partial \tilde{\boldsymbol{w}}} \bigg|_{\tilde{\boldsymbol{w}} = \tilde{\boldsymbol{w}}_0} \right\| \cdot \|\tilde{\boldsymbol{w}}^* - \tilde{\boldsymbol{w}}_0\| + O(\|\tilde{\boldsymbol{w}} - \tilde{\boldsymbol{w}}_0\|^2). \end{aligned}$$

Invoking Theorem D.1, we know:

$$\lim_{n \to \infty} \|\tilde{\boldsymbol{w}}^* - \tilde{\boldsymbol{w}}_0\| \xrightarrow{P} 0,$$

which implies:

$$\lim_{n \to \infty} \|\boldsymbol{w}^* - \boldsymbol{w}_0\| \xrightarrow{P} 0.$$

$\square$

# E  Experiments Appendix

## E.1  Computing Platform

All simulations are implemented using MATLAB R2021b on a PC running Windows 10 Intel(R) Xeon(R) E5-2650 v4 CPU (2.2GHz) and 64GB RAM. We report the results of synthetic datasets in the main paper and defer other results to the supplementary material.

## E.2 More experiments

Experimental results for $\frac{d}{n} = \frac{1}{3}$ are displayed in Figure F1. As shown in Figure F1, the estimation error $\|\boldsymbol{w}^* - \boldsymbol{w}_0\|$ generated by the SCLS method decreases to $0$ as $n$ goes to $\infty$. This trend is consistent with the conclusion from Figure 1, Theorems 4.3 and D.2, thereby affirming the efficacy and reliability of the SCLS method.

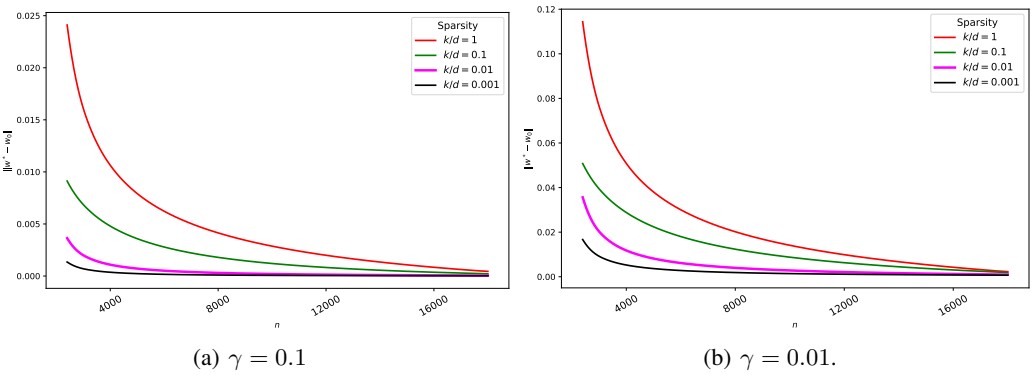

(a) $\gamma = 0.1$      (b) $\gamma = 0.01$.

Figure F1: The change of $\|\boldsymbol{w}^* - \boldsymbol{w}_0\|$ with $n$ for SCLS method under different Sparsity $k/d$.

