# OpenReview forum: "Error Analysis of Spherically Constrained Least Squares Reformulation in Solving the Stackelberg Prediction Game"
_NeurIPS.cc/2024/Conference — NeurIPS 2024 poster_

### Official Review · Reviewer_XvKF · 2024-07-10

**Soundness:** 3
**Presentation:** 3
**Contribution:** 4
**Rating:** 8
**Confidence:** 4

**Summary:**

The Stackelberg Prediction Game (SPG) models with the least squares loss variant (SPG-LS) gaining attention. The spherically constrained least squares (SCLS) method is the latest state-of-the-art method for solving the SPG-LS problem. The authors address the lack of theoretical error analysis for the SCLS method. By transforming the estimation error into an Auxiliary Optimization (AO) problem using the Convex Gaussian Min-Max Theorem (CGMT), the authors provide a theoretical error analysis, confirming the SCLS method's reliability. Experimental results validate the theorems.

**Strengths:**

The authors focus on the Spherically Constrained Least Squares (SCLS) method which is the state-of-the-art method for solving the SPG-LS problem. The lack of theoretical error analysis of the SCLS method restricts its large-scale applications. The authors investigate the estimation error between the learner obtained by the SCLS method and the actual learner. Therefore, the content of this paper is original. The writing quality is high, and the argumentation process is clear. Moreover, this analysis strengthens the theoretical framework of the SCLS method and confirms the reliability of the learner produced by the SCLS method. Thus, the authors’ contribution is significant in promoting the large-scale applications of the SCLS method.

**Weaknesses:**

1. The original paper of the SCLS method focuses on optimization (9), but the authors explore the SCLS problem (6). What are the differences between (6) and (9)? Why can the authors replace optimization (9) with (6)?

2. How do the authors derive optimization (32) from (31)?

3. There is a remark for Theorem 3.2, however, Theorem 3.1 lacks explanation.  The authors should add explanatory content to Theorem 3.1 to improve its readability.

4. Theorem 3.1 appears very similar to Theorem 3.2. What’s the relationship between Theorem 3.1 and Theorem 3.2?

**Questions:**

See the Weaknesses.

**Limitations:**

In the Appendix, the limitations are discussed.

---

> ### Author Rebuttal · Authors · 2024-08-04
>
> ## Answer to Reviewer XvKF
>
> Dear Reviewer XvKF,
>
> Thank you for your job in reviewing our paper. We are very sorry for the inconvenience caused by our presentations. To this end, following your comments, we correct our work in the revision. All the references appear in our main paper.
>
> In regards to the weaknesses:
>
> __Weakness 1.__ The original paper of the SCLS method focuses on optimization (9), but the authors explore the SCLS problem (6). What are the differences between (6) and (9)? Why can the authors replace optimization (9) with (6)?
>
> __Answer:__  Thank you for your comments. The SCLS problems (6) is equivalent to optimization (9).
>
> $$\begin{align*}
> \min _{\pmb{r}}q(\pmb{r}), \quad \text{s.t.}\quad \pmb{r}^T\pmb{r} = 1,\tag{9}
> \end{align*}$$
>
> where $ q(\pmb{r}) = ||\hat{L}\pmb{r}-(\pmb{y}-\pmb{z}/2)||^2$, $\hat{L} = \begin{pmatrix}\frac{\sqrt{\gamma}}{2}\pmb{X}&\frac{\pmb{z}}{2}\end{pmatrix}$ and $\pmb{r}^\top=\begin{pmatrix}
> \tilde{\pmb{w}}\\ \tilde{\alpha}
> \end{pmatrix}.$ Then,
>
> $$
> \begin{align*}
> q(\pmb{r})&= ||\hat{L}\pmb{r}-(\pmb{y}-\pmb{z}/2)||^2= ||\begin{pmatrix}\frac{\sqrt{\gamma}}{2}\pmb{X}&\frac{\pmb{z}}{2}\end{pmatrix}\begin{pmatrix}
> \tilde{\pmb{w}}\\ \tilde{\alpha}
> \end{pmatrix}^\top-(\pmb{y}-\pmb{z}/2) ||^2\\\\
> &=\Vert \frac{\tilde \alpha}{2}  \pmb{z} + \frac{\sqrt{\gamma}}{2}\pmb{X}\tilde{\pmb{w}} - (\pmb{y} - \frac{\pmb{z}}{2})\Big\Vert^2.
> \end{align*}
> $$
> Additionally, the constraint can be transformed to
> $$
> \begin{align*}
> 1=\pmb{r}^T\pmb{r}=\begin{pmatrix}
> \tilde{\pmb{w}}\\ \tilde{\alpha}
> \end{pmatrix}^\top\begin{pmatrix}
> \tilde{\pmb{w}}\\ \tilde{\alpha}
> \end{pmatrix}=\tilde{\pmb{w}}^\top\tilde{\pmb{w}} + \tilde \alpha^2.
> \end{align*}
> $$
> Therefore, the optimization (9)  is equivalent to SCLS problems (6):
> $$\begin{align}
>     \min\limits _{\tilde{\pmb{w}}, \tilde{\alpha}}\quad  {\tilde v}(\tilde{\pmb{w}},\tilde \alpha) \triangleq \Big\Vert \frac{\tilde \alpha}{2}  \pmb{z} + \frac{\sqrt{\gamma}}{2}\pmb{X}\tilde{\pmb{w}} - (\pmb{y} - \frac{\pmb{z}}{2})\Big\Vert^2,\quad
>     {\text{s.t.} }\; \tilde{\pmb{w}}^\top\tilde{\pmb{w}} + \tilde \alpha^2 = 1. \tag{6}
> \end{align}$$
>
>
> __Weakness 2.__ How do the authors derive optimization (32) from (31)?
>
> __Answer:__  Thank you for your comments. We are very sorry for the inconvenience caused by our presentations. The detailed derivation process from (31) to (32) is:
> $$\begin{align*}
>     \min _{\tilde{\pmb{\beta}}}\max _{\eta \geq 0}
>    \frac{1}{n}\Big(\sqrt{\Vert\tilde{\pmb{\beta}}\Vert^2+(\pmb{c}^\top\tilde{\pmb{\beta}})^2+\frac{4\sigma^2}{\gamma}}\cdot\Vert\pmb{g}\Vert \eta
>     + \eta\pmb{h}^\top\tilde{\pmb{\beta}}
>     -\frac{\eta^2}{4}\Big). \tag{31}
> \end{align*}$$
> The formulation (31) is a quadratic function of $\eta$ with the symmetric axis:
> $$\begin{align*}
>     \eta _s=2\Big(\sqrt{\Vert\tilde{\pmb{\beta}}\Vert^2+(\pmb{c}^\top\tilde{\pmb{\beta}})^2+\frac{4\sigma^2}{\gamma}}\cdot\Vert\pmb{g}\Vert+\pmb{h}^\top\tilde{\pmb{\beta}}\Big)
>     >\Vert\tilde{\pmb{\beta}}\Vert(\Vert\pmb{g}\Vert-\Vert\pmb{h}\Vert).
> \end{align*}$$
>
> Additionally, $\eta _s(\Vert\pmb{g}\Vert+\Vert\pmb{h}\Vert)>\Vert\tilde{\pmb{\beta}}\Vert(\Vert\pmb{g}\Vert^2-\Vert\pmb{h}\Vert^2)$.  $\Vert\pmb{g}\Vert^2$ and $\Vert\pmb{h}\Vert^2$ concentrate around their means $n$ and $d$, respectively.  Consequently, the value around which $\eta _s$ concentrates is nonnegative, due to $d/n<1$.  Moreover, taking $\eta _s$ into (31):
> $$\begin{align*}
>     &\min _{\tilde{\pmb{\beta}}}
>     \frac{1}{n}\Big(\sqrt{\Vert\tilde{\pmb{\beta}}\Vert^2+(\pmb{c}^\top\tilde{\pmb{\beta}})^2+\frac{4\sigma^2}{\gamma}}\cdot\Vert\pmb{g}\Vert
>     + \pmb{h}^\top\tilde{\pmb{\beta}}
>     \Big )^2\\\\
>     =& \min _{\tilde{\pmb{\beta}}}
>     \frac{1}{n}\Big[(\Vert\tilde{\pmb{\beta}}\Vert^2+(\pmb{c}^\top\tilde{\pmb{\beta}})^2+\frac{4\sigma^2}{\gamma})\Vert\pmb{g}\Vert^2+ (\pmb{h}^\top\tilde{\pmb{\beta}})^2+2\pmb{h}^\top\tilde{\pmb{\beta}}\Vert\pmb{g}\Vert
>     \sqrt{\Vert\tilde{\pmb{\beta}}\Vert^2+(\pmb{c}^\top\tilde{\pmb{\beta}})^2+\frac{4\sigma^2}{\gamma}}\Big].\tag{32}
> \end{align*}$$
>
> (Please refer to lines 221-228 of our main paper.)
>
> __Weakness 3.__ There is a remark for Theorem 3.2, however, Theorem 3.1 lacks explanation. The authors should add explanatory content to Theorem 3.1 to improve its readability.
>
> __Answer:__  Thank you for your suggestion. Following your recommendation, we will add the following remark for Theorem 3.1 in our revision to explain its content:
>
> Remark: Theorem 3.1 indicates that, as $n$ goes to $\infty$, the parameter vector $\tilde{\pmb{w}}^*$ learned through the SCLS method (6) reliably converges to actual parameter vector $\tilde{\pmb{w}} _0$ in probability.
> We then can utilize Theorem 2.4 to establish the estimation error of SPG-LS (1) solved by the SCLS (6).
>
> __Weakness 4.__ Theorem 3.1 appears very similar to Theorem 3.2. What’s the relationship between Theorem 3.1 and Theorem 3.2?
>
> __Answer:__  Thank you for your comments. Theorem 3.1 indicates that
> $$\begin{align*}
>     \lim _{n\to \infty}\Vert\tilde{\pmb{w}}^*-\tilde{\pmb{w}} _0\Vert \stackrel{P}{\longrightarrow} 0,
> \end{align*}$$
> which corresponds to the estimation error analysis of the SCLS problem (6). Theorem 3.2 demonstrates that
> $$\begin{align*}
>     \lim _{n \to \infty}\Vert\pmb{w}^*-\pmb{w} _0\Vert \stackrel{P}{\longrightarrow} 0,
> \end{align*}$$
> which corresponds to the estimation error of SPG-LS (1) recovered from the SCLS problem (6). Therefore, Theorem 3.1 is different from Theorem 3.2, and Theorem 3.1 can be used to derive Theorem 3.2.

---

> > ### Comment · Reviewer_XvKF · 2024-08-09
> >
> > Thanks the authors for the reply. They have addressed my questions well. I am pleased to keep my positive score.

---

> > > ### Author Response · Authors · 2024-08-11
> > > **Appreciation to Reviewer XvKF**
> > >
> > > Thank you very much for your acknowledgment and efforts.

---

### Official Review · Reviewer_PFXk · 2024-07-11

**Soundness:** 3
**Presentation:** 2
**Contribution:** 2
**Rating:** 5
**Confidence:** 3

**Summary:**

The paper investigates the estimation error of the learner obtained through the SCLS method proposed in [3] in comparison to the actual learner. It redefines the estimation error of the SCLS method as a Primary Optimization (PO) problem and applies the Convex Gaussian min-max theorem (CGMT) to convert the PO problem into an Auxiliary Optimization (AO) problem. Subsequently, a theoretical error analysis for the SCLS method is presented based on this simplified AO problem. This analysis validates the accuracy of the strategy generated by the SCLS method from [3].

**Strengths:**

The paper introduces a method to analyze the error of SCLS. And this method is applicable to other statistical learning algorithms.

**Weaknesses:**

1. The reason for ``the lack of theoretical analysis on the error of the SCLS method limits its large-scale applications'' is unclear, which is very important to the motivation of this paper. Therefore, I failed to understand the importance of studying this problem, as it appears to be artificially created.

2. The notations in this article are somewhat ambiguous, hindering readability. For instance, the notations $d\in \mathbb{N}$ and $n=n(d)$ in Definition 2.5 (GMT admissible sequence) are identical to the notation for the data dimension $X\in \mathbb{R}^{n\times d}$.

3. The assumption ``The data $X$ is drawn i.i.d. from $N(0,1)$'' appears to be overly restrictive within the framework of the Stackelberg prediction game.

4. Section 2.1 of this paper is redundant as all the relevant information is extracted from reference [3].

**Questions:**

1. Line 61: What is the main distinction between $w_0$ and $w^*$?
2. Line 114: The notation ($\omega,~\alpha$) appears twice.
3. Line 155: The authors state that "a challenging PO problem can be replaced with a simplified AO problem." Could the authors provide more explanation on where the challenge in the PO problem lies and why the AO problem is simpler? Additionally, on page 7, line 213, why is PO (28) more complex than AO (29)? Where does the difficulty in (28) lie?
4. Line 161: $(u)$ should be changed to $f^*(u)$.
5. Line 170: The authors should explain why the second equation in (16) is valid.
6. Line 178: I am curious as to why the variables $\tilde{w_0}$ and $\tilde{\alpha_0}$ are not included as variables in the objective function of problem (21).
7. Line 202: As the authors state, "This equivalence allows for the substitution of the analysis of the optimal cost $\tilde{\beta}^*$ in SCLS (21) with the analysis of the optimal solution $\hat{\beta}^*$ in optimization (26)." We can see that (21) is nonconvex, while (26) is an unconstrained convex optimization. Does this mean the authors have equivalently reformulated a nonconvex optimization problem into a convex optimization?
8. It is reasonable to consider whether the limit of $\lim_{n\rightarrow \infty}\frac{d}{n}$ falls within the interval $[(0,1)]$ since this assumption is not adopted in [3]. Furthermore, it is impractical to assume that $d/n < 1$, as the features of the data may surpass the sample size.
9. In experiments, why the true parameter $w_0$ is generated randomly?

---

> ### Author Rebuttal · Authors · 2024-08-04
>
> Dear Reviewer PFXk, Thank you for the detailed and thorough review. All the references can be seen in Author Rebuttal by Authors.
>
> ### For Weaknesses:
>
> __W1:__ The SCLS method [3] is the latest advanced technique for solving the widely applicable SPG-LS problem, winning the ICML 2022 Outstanding Paper Award. However, the SCLS method lacks error analysis, and we have filled this gap. Error analysis of algorithms is a very crucial and popular topic in the field of machine learning, as evidenced by several papers [B1-B8]. Therefore, it is important to study the error of the SCLS algorithm.
>
> We will clarify this sentence in the revision as follows: The paper [3] lacks theoretical analysis on the error of the SCLS method.
>
> __W2:__ In our paper, $d$ denotes the sample dimension and $n$ is the number of samples. Thus, $d$ in Definition 2.5 and $d$ in $X\in R^{n\times d}$ both represent the sample dimension, meanwhile, $n$ in Definition 2.5 and $n$ in $X\in R^{n\times d}$ both denote the number of samples, ensuring consistency and clarity. (See lines 31-33)
>
> __W3:__ The SCLS method is currently the state-of-the-art for solving SPG-LS, having won the ICML 2022 Outstanding Paper Award [3]. However, [3] lacks theoretical analysis on the error of the SCLS method. To the best of our knowledge, we are the first to investigate the error of the SCLS method. The primary contribution of our paper is to provide a theoretical perspective on the error of the SCLS method under Gaussian assumption. Our limitation section highlights this Gaussian assumption (See lines 372-375). Reviewer btrD also identifies this limitation but acknowledges our contribution under Gaussian settings.
>
> It is worth emphasizing that the Gaussian hypothesis is a commonly used approach for theoretical analysis of algorithms in machine learning, as evidenced by papers  [C1~C8]. Thus, we investigate the error of the SCLS algorithm under Gaussian settings.
>
> __W4:__ Since our aim is to study the error of the SCLS method, it is important to provide an overview of this method. Section 2.1 is the preliminaries to introduce the necessary concepts and theorems about the SCLS method from [3]. These concepts and theorems are frequently used and play a crucial role in the subsequent derivation of our manuscript. Following your advice, we will simplify Section 2.1 in our revision.
>
> ### For Questions:
>
> __Q1:__  $w_0$ is the known parameter of the SPG-LS model (1), while $w^*$ learned by the SCLS method is an estimator of $w_0$. (See lines 61-66)
>
> __Q2:__  We will rectify it in revision.
>
> __Q3:__ PO problem contains the matrix $G\in R^{n\times d}$ and the challenge lies in the processing of matrices. The inclusion of a matrix in the PO problem increases the complexity of analysis. In contrast, the AO problem only contains vectors with dimensions $d$ or $n$, where vectors are easier to handle than matrices. Additionally, the AO problem reduces the dimension of the PO problem from $n\times d$ to max{$d, n$}, thereby simplifying the PO problem.
>
> According to the analysis of the PO and AO problems above, the reasons why PO (28) is  more complex than AO (29) and the difficulties of (28) are summarized as follows:
>
> (i). The PO (28) contains the matrix $X\in R^{n\times d}$ and the challenge lies in processing matrices. the AO (29) only contains vectors with dimensions $d$ or $n$, which are easier to handle than matrices.
>
> (ii). The AO (29) reduces the dimension of the PO (28) from $n\times d$ to max{$d, n$}, thereby simplifying the PO (28).
>
> (iii). It is difficult to obtain the value that the PO (28) concentrates on.
>
> (iv). The AO problem (29) can be further simplified by AO optimization (33) that only includes estimation error variable $\tilde{\beta}$, which is easier to analyze than PO (28).
>
> __Q4:__ We will rectify it in revision.
>
> __Q5:__ The SPG-LS model (1) can be expressed as :
> $$\min_ w||X^ *w-y||^2,s.t.X^ * =argmin_ {\hat X}||\hat Xw-z||^2+\gamma|| \hat X-X||_ F^2.\tag{1}$$
>
> [1, 2, 3] reformulate SPG-LS (1) into :
> $$\inf_ {w}\Big\Vert\frac{\frac{1}{\gamma}zw^Tw+Xw}{1+\frac{1}{\gamma}w^Tw}-y\Big\Vert^2,\tag{4}$$
> which indicates that
> $$X^*w = \frac{\frac{1}{\gamma}zw^Tw+Xw}{1+\frac{1}{\gamma}w^Tw}.$$
>
> Therefore, if $w=w _0$, we have
> $$y=X^*w _0+\epsilon=\frac{\frac{1}{\gamma}zw _0^Tw _0+Xw _0}{1+\frac{1}{\gamma}w _0^Tw _0}+\epsilon=\frac{\alpha _0z+Xw _0}{1+\alpha _0}+\epsilon,\tag{16}$$
> where $\alpha_0=w_0^Tw_0/\gamma$. (See lines 92-95 and 169-170)
>
> __Q6:__ According to our answer to Q1, $w_0$ is assumed to be known. Since $\alpha_0=w_0^Tw_0/\gamma$, both $\tilde{w}_0$ and $\tilde{\alpha}_0$ are known constants, not variables.
>
> __Q7:__ Following the approach of [D1-D6], (26) is the first-order approximation of (21) where the derivation process is in lines 183-199. We will clarify this point in our revision.
>
> __Q8:__ To the best of our knowledge, we are the first to investigate the error of the SCLS method, and the theoretical analysis in our paper successfully explains the behavior of the SCLS method when $d/n<1$. Thus, the primary contribution of our paper is to provide a theoretical perspective on the error of the SCLS method under certain conditions. Analyzing the error of the SCLS method under broad conditions is beyond the scope of this paper. We plan to investigate the error results of the SCLS method for $d/n>1$ in our future work.
>
> __Q9:__ According to our answer to Q1, $w_0$ represents the ``true'' weight parameter of the SPG-LS model, which is assumed to be known. Therefore, once selected, $w_0$ should be fixed in experiments. In addition to random generation, we can also manually set a constant value for $w_0$.
>
> We sincerely thank you once again for your time and effort in reviewing our paper. We hope that our answers have met your expectations and satisfaction.

---

> > ### Comment · Reviewer_PFXk · 2024-08-11
> > **responce**
> >
> > I would like to thank the authors for their responses. While I may not be fully familiar with the topic of error analysis, I remain uncertain about the sufficiency of the authors' contribution. However, after considering the feedback from other reviewers, I have decided to raise the score to 5.

---

> > > ### Author Response · Authors · 2024-08-11
> > > **Appreciation for Raising Score**
> > >
> > > We appreciate your decision to raise our score.  We will further emphasize our contribution  in the revision.

---

> ### Author Response · Authors · 2024-08-10
> **Kindly Requesting Confirmation on Responses**
>
> Dear Reviewer  PFXk,
>
> We hope this message finds you well. I am reaching out to kindly request your prompt response to confirm whether our responses adequately address your queries. We sincerely thank you for your time and effort during this discussion period. Your timely feedback is greatly appreciated.

---

### Official Review · Reviewer_btrD · 2024-07-11

**Soundness:** 3
**Presentation:** 3
**Contribution:** 3
**Rating:** 7
**Confidence:** 4

**Summary:**

The spherically constrained least squares reformulation method proposed by Jiali et al. has shown superior performance in addressing the issues of the Stackelberg prediction game. This paper aims to analyze the error between the estimators and the ground truth. The main theory shows that the estimation error approaches to zero in probability. The empirical studies verify the claims appear in the paper.

**Strengths:**

- This paper studied the seminal work of SPGs published in ICML 2022. They are the first to conduct the estimation error analysis.
- Technically, they first reformulate the estimation error of the SCLS method as a PO problem. Then, it is novel to use the CGMT to reduce the PO problem into the AO problem. Some derivations are performed to further simplify the AO problem, as shown in Eq. (33). They finally analyze the simple Eq. (33) and present two asymptotic main theories. The empirical studies are also conducted to verify the theory. Overall, the presentation is clear and the paper seems to be a theoretical solid paper.

**Weaknesses:**

- As pointed by the author in the checklist, the main weakness is that this paper assumes the Gaussian inputs. This paper may motivate further research to remove or weaken the assumptions.
- Eq.(13) and Eq.(14) miss the transpose symbol $T$.
- Some derivations need to be more clear, for example, in Line 211, how can the author derive the Eq.(29)?
- The asymptotic results of CGMT relies on $d$, while Theorems 3.1 and 3.2 holds when $\lim_{n\to infty}$. Could the authors say something about this?

**Questions:**

- How can the authors derive Eq.(29)?
- Could the authors specify the feasibility of replacing $d$ required by CGMT with $n$ in their conclusions.

**Limitations:**

Yes

---

> ### Author Rebuttal · Authors · 2024-08-04
>
> ## Answer to Reviewer  btrD
>
> Dear Reviewer btrD,
>
> Thank you for your job in reviewing our paper. We are very sorry for the inconvenience caused by our presentations. We extend our heartfelt gratitude for your patience and meticulous guidance. Your insightful comments is valuable for us and we appreciate the opportunity to address your concerns.
>
> ### In regards to the weaknesses:
>
> __Weakness 1.__  As pointed by the author in the checklist, the main weakness is that this paper assumes the Gaussian inputs. This paper may motivate further research to remove or weaken the assumptions.
>
> __Answer:__
>
> Thank you for your comments.  To the best of our knowledge, we are the first to investigate the error of the SCLS method. The primary contribution of our paper is to provide a theoretical perspective on the error of the SCLS method under the Gaussian assumption. It is worth emphasizing that the Gaussian hypothesis is a commonly used approach for theoretical analysis in the field of machine learning, as evidenced by papers [A1~A8]. The error analysis of the SCLS method under general or weaker conditions is beyond the scope of this paper. We will investigate the results of non-Gaussian settings in our future work.
>
> [A1]. Alexander Camuto, Matthew Willetts, Umut Simsekli, Stephen J. Roberts, Chris C. Holmes: Explicit Regularisation in Gaussian Noise Injections. NeurIPS 2020
>
> [A2]. Prathamesh Mayekar, Jonathan Scarlett, Vincent Y. F. Tan:
> Communication-Constrained Bandits under Additive Gaussian Noise. ICML 2023: 24236-24250
>
> [A3]. Matthew Joseph, Alexander Yu: Some Constructions of Private, Efficient, and Optimal K-Norm and Elliptic Gaussian Noise. COLT 2024: 2723-2766
>
> [A4]. Alexander Camuto, Xiaoyu Wang, Lingjiong Zhu, Chris C. Holmes, Mert Gürbüzbalaban, Umut Simsekli: Asymmetric Heavy Tails and Implicit Bias in Gaussian Noise Injections. ICML 2021: 1249-1260
>
> [A5]. Holden Lee, Chirag Pabbaraju, Anish Prasad Sevekari, Andrej Risteski:
> Pitfalls of Gaussians as a noise distribution in NCE. ICLR 2023
>
> [A6]. Christos Thrampoulidis, Ehsan Abbasi, Babak Hassibi: Precise Error Analysis of Regularized M-Estimators in High Dimensions. IEEE Trans. Inf. Theory 64(8): 5592-5628 (2018)
>
> [A7]. Yufeng Zhang, Jialu Pan, Li Ken Li, Wanwei Liu, Zhenbang Chen, Xinwang Liu, Ji Wang: On the Properties of Kullback-Leibler Divergence Between Multivariate Gaussian Distributions. NeurIPS 2023
>
> [A8]. Seunghyuk Cho, Juyong Lee, Dongwoo Kim: Hyperbolic VAE via Latent Gaussian Distributions. NeurIPS 2023
>
> __Weakness 2.__ Eq.(13) and Eq.(14) miss the transpose symbol $T$.
>
> __Answer:__  Thank you for your comments. We are very sorry for this clerical error and we will add  the transpose symbol $T$ to rectify Equations (13) and (14) in our revision.
>
> __Weakness 3.__ Some derivations need to be more clear, for example, in Line 211, how can the author derive the Eq.(29)?
>
> __Answer:__  Thank you for your comments.
> $$
> \Phi _{\text{SCLS}}(\pmb{X})= \min _{\tilde{\pmb{\beta}}} \max _{\pmb{u}}\frac{1}{n}\big(
> \pmb{u}^\top\pmb{X}\tilde{\pmb{\beta}}+\psi(\tilde{\pmb{\beta}}, \pmb{u})\big),\tag{28}
> $$
> where
> $\psi(\tilde{\pmb{\beta}}, \pmb{u}):=\pmb{c}^\top\tilde{\pmb{\beta}}\cdot\pmb{u}^\top \pmb{z}-\frac{2\pmb{u}^\top\pmb{\epsilon}}{\sqrt{\gamma}}-\frac{\Vert\pmb{u}\Vert^2}{4}.$
> Given that the entries of $\pmb{X}$ are drawn i.i.d. from $\mathcal{N}(0, 1)$ and $\psi(\tilde{\pmb{\beta}}, \pmb{u})$ is a convex-concave function, the $\textbf{PO}$ problem (28) satisfies the conditions of Theorem 2.6. Consequently, we replace the challenging $\textbf{PO}$ problem (28) with a simplified $\textbf{AO}$ problem using CGMT:
>
> $$\begin{align*}
>     \phi _{\text{SCLS}}(\pmb{g},\pmb{h}) =& \min _{\tilde{\pmb{\beta}}} \max _{\pmb{u}}\frac{1}{n}\big(\Vert\tilde{\pmb{\beta}}\Vert\pmb{g}^\top\pmb{u}
>     + \Vert\pmb{u}\Vert\pmb{h}^\top\tilde{\pmb{\beta}}
>      +\pmb{c}^\top\tilde{\pmb{\beta}}\cdot\pmb{u}^\top \pmb{z}-\frac{2\pmb{u}^\top\pmb{\epsilon}}{\sqrt{\gamma}}-\frac{\Vert\pmb{u}\Vert^2}{4}\big)\\\\
>     =&\min _{\tilde{\pmb{\beta}}} \max _{\pmb{u}}\frac{1}{n}\big[(\Vert\tilde{\pmb{\beta}}\Vert\pmb{g}+\pmb{c}^\top\tilde{\pmb{\beta}}\pmb{z}-\frac{2\pmb{\epsilon}}{\sqrt{\gamma}})^\top\pmb{u}
>     + \Vert\pmb{u}\Vert\pmb{h}^\top\tilde{\pmb{\beta}}-\frac{\Vert\pmb{u}\Vert^2}{4}\big],\tag{29}
> \end{align*}$$
>
> (Please refer to lines 129-131 and 207-212 of our main paper.)
>
> __Weakness 4.__  The asymptotic results of CGMT relies on $d$, while Theorems 3.1 and 3.2 holds when $\lim _{n\to \infty}$. Could the authors say something about this?
>
> __Answer:__  Thank you for your comments. Definition 2.5 indicates that $n=n(d)$ where $n(d)$ is a funtion of $d$. Additionally, Theorems 3.1 and 3.2 require $\lim _{n\to\infty}\frac{d}{n}\in (0,1)$. Therefore, $\lim _{n\to infty}$ implies $\lim _{d\to \infty}$. The transformation from $n$ to $d$ doesn't affect our theoretical results. It is worth emphasizing that all deductions should be based on $d$, if we replace $n$ with $d$.
>
> ### In regards to your questions:
>
> __Question 1.__ How can the authors derive Eq.(29)?
>
> __Answer:__  Thank you for your comments. Question 1 is similar to Weakness 3. Please see the Answer to Weakness 3.
>
> __Question 2.__ Could the authors specify the feasibility of replacing $d$ required by CGMT with $n$ in their conclusions.
>
> __Answer:__  Thank you for your comments. Question 2 is similar to Weakness 4. Please see the Answer to Weakness 4.

---

> > ### Comment · Reviewer_btrD · 2024-08-14
> >
> > Thank you for the author's response. My issue has been resolved, and I will maintain my score.

---

> > > ### Author Response · Authors · 2024-08-14
> > >
> > > Thank you very much for your acknowledgment and efforts.

---

### Official Review · Reviewer_rMtE · 2024-07-12

**Soundness:** 4
**Presentation:** 3
**Contribution:** 4
**Rating:** 8
**Confidence:** 4

**Summary:**

The spherically constrained least squares reformulation (SCLS) method proposed in paper [3] is the state-of-the-art method for solving the Stackelberg prediction game with least squares loss (SPG-LS), and the paper [3] has won the ICML 2022 Outstanding Paper Award. This paper further enhances the theoretical framework of the SCLS method by providing a theoretical error analysis using the Convex Gaussian Min-Max Theorem (CGMT). The theoretical results indicate that the learner obtained through the SCLS method reliably converges to the actual learner vector of the Stackelberg prediction game. Additionally, the authors conduct experiments to validate their theorems, with the experimental results aligning with their theoretical predictions.


[3] Jiali Wang, Wen Huang, Rujun Jiang, Xudong Li, and Alex L. Wang. Solving Stackelberg prediction game with least squares loss via spherically constrained least squares reformulation. In ICML, volume 162, pages 22665–22679, 2022.

**Strengths:**

$\bullet$ __Originality__

1. This paper provides a theoretical error analysis of the SCLS method proposed by the paper `` Solving Stackelberg prediction game with least squares loss via spherically constrained least squares reformulation ’’, which won the ICML 2022 Outstanding Paper Award.

2. This paper introduces the transformation from the Primary Optimization (PO) problem to the Auxiliary Optimization (AO) problem.

3. This paper applies the Convex Gaussian min-max theorem.

$\bullet$ __Quality__ The quality of this paper is good.

$\bullet$ __Clarity__  The overall expression and proof process of this paper are very clear.

$\bullet$ __Significance__  The theoretical results further ensure the effectiveness and accuracy of the SCLS method proposed by [3]. This topic is interesting and important.

**Weaknesses:**

$\bullet$ This paper transforms a PO problem related to the error of the SCLS method into a simplified AO problem, but the motivation for this transformation seems weak. More specific reasons for this transformation are needed.

$\bullet$ According to Theorem 2.6, the equation between lines 154-155 should be a result of asymptotic convergence. However, the condition ``$n \to \infty$’’ is missing in the description.


$\bullet$ According to (2), the authors translate the findings of the approximated SCLS problem to that of the original SCLS problem if $\Vert\tilde{w}-\tilde{w}_0\Vert\to 0$ happens independently. Which place reflects that this condition will occur?

**Questions:**

$\bullet$ None.

**Limitations:**

$\bullet$ Yes, the authors have adequately addressed the limitations.

---

> ### Author Rebuttal · Authors · 2024-08-04
>
> ## Answer to Reviewer rMtE
>
> Dear Reviewer rMtE,
>
> Thank you for your job in reviewing our paper. We are very sorry for the inconvenience caused by our presentations. To this end, following your comments, we correct our work in the revision.
>
> In regards to the weaknesses:
>
> __Weakness 1.__ This paper transforms a PO problem related to the error of the SCLS method into a simplified AO problem, but the motivation for this transformation seems weak. More specific reasons for this transformation are needed.
>
> __Answer:__  Thank you for your comments. Theorem 2.6 indicates that, if the optimal cost $\phi(\pmb{g},\pmb{h})$ of $\textbf{AO}$ concentrates to some value $\mu$, the same holds true for $\Phi(\pmb{G})$ of $\textbf{PO}$. Furthermore, under appropriate additional assumptions, the optimal solutions of the $\textbf{AO}$ and $\textbf{PO}$ problems are also closely related by $\Vert\pmb{\beta}  _{\Phi}(\pmb{G})\Vert = \Vert \pmb{\beta} _{\phi}(\pmb{g},\pmb{h})\Vert$, as $n\to \infty$. This suggests that, within the CGMT framework, a challenging $\textbf{PO}$ problem can be replaced with a simplified $\textbf{AO}$ problem, from which the optimal solution of the $\textbf{PO}$ problem can be accurately inferred. Moreover, we are more concerned about $\Vert \pmb{\beta} _{\phi}(\pmb{g},\pmb{h})\Vert$. Therefore, if we reduce the AO problem only involves scalar variable about $\Vert \pmb{\beta} _{\phi}(\pmb{g},\pmb{h})\Vert$, we obtain the error of the SCLS method by the relationship $\Vert\pmb{\beta} _{\Phi}(\pmb{G})\Vert = \Vert \pmb{\beta} _{\phi}(\pmb{g},\pmb{h})\Vert$.
>
> If the optimal solution of optimization (33) is $\Vert\tilde{\pmb{\beta}}\Vert=\rho^*$, we have $\Vert\tilde{\pmb{\beta}} _{\phi _{\text{SCLS}}}\Vert\stackrel{P}{\longrightarrow}\rho^*$ for $\textbf{AO}$ problem (29). Then, by virtue of CGMT, $\Vert\tilde{\pmb{\beta}} _{\Phi _{\text{SCLS}}}\Vert\stackrel{P}{\longrightarrow}\rho^*$ also holds for $\textbf{PO}$ problem (28). If $\rho^*$ further satisfies $\rho^* = 0$, based on the relationship between the original and approximated SCLS in Section 3.1, we have $\Vert\tilde{\pmb{w}}-\tilde{\pmb{w}} _0\Vert\stackrel{P}{\longrightarrow} 0$ for SCLS problems (21) and (6). Therefore, it only remains to obtain the optimal value of $\rho$ in optimization (33) that plays the role of $\Vert\tilde{\pmb{\beta}}\Vert$.
>
> (Please refer to lines 152-157 and 236-241 of our main paper.)
>
> __Weakness 2.__ According to Theorem 2.6, the equation between lines 154-155 should be a result of asymptotic convergence. However, the condition ``$n \to \infty$’’ is missing in the description.
>
> __Answer:__  Thank you for your comments. The equation between lines 154-155 is a further explanation of Theorem 2.6 and the condition ``$n \to \infty$’’ is included in Theorem 2.6.
>
> __Weakness 3.__ According to (2), the authors translate the findings of the approximated SCLS problem to that of the original SCLS problem if $\Vert\tilde{w}-\tilde{w} _0\Vert\to 0$ happens independently. Which place reflects that this condition will occur?
>
> __Answer:__ Thank you for your comments. Our Theorem 3.1 reflects that this condition happens independently. Specifically, Theorem 3.1 demonstrates that $\lim _{n\to \infty}\Vert\tilde{\pmb{w}}^*-\tilde{\pmb{w}} _0\Vert \stackrel{P}{\longrightarrow} 0$ for the approximated SCLS problem. Then, according to the relationship (25),  we can translate the findings of the approximated SCLS problem to that of the original SCLS problem.

---

> > ### Comment · Reviewer_rMtE · 2024-08-10
> > **response**
> >
> > The author did a good job in answering my questions. I am happy to recommend the acceptance.

---

> > > ### Author Response · Authors · 2024-08-11
> > > **Appreciation to Reviewer rMtE**
> > >
> > > We deeply appreciate your acknowledgment, praise, and efforts.

---

### Author Rebuttal · Authors · 2024-08-04

## The references mentioned in the Rebuttal to Reviewer PFXk.

__For Weakness 1.__

[B1] Estimating the Error of Randomized Newton Methods: A Bootstrap Approach. ICML 2020

[B2] $\ell _{1, p}$-Norm Regularization: Error Bounds and Convergence Rate Analysis of First-Order Methods. ICML 2015

[B3] Addressing Function Approximation Error in Actor-Critic Methods. ICML 2018

[B4] Faster Algorithms and Constant Lower Bounds for the Worst-Case Expected Error. NeurIPS 2021

[B5] A Comparison of Hamming Errors of Representative Variable Selection Methods. ICLR 2022

[B6] On Generalization Error Bounds of Noisy Gradient Methods for Non-Convex Learning. ICLR 2020

[B7] Generalization error of spectral algorithms. ICLR 2024

[B8] Error Estimation for Randomized Least-Squares Algorithms via the Bootstrap. ICML 2018

__For Weakness 3.__

[C1] Near-Optimal Algorithms for Gaussians with Huber Contamination: Mean Estimation and Linear Regression. NeurIPS 2023

[C2] Differentially Private Algorithms for Learning Mixtures of Separated Gaussians. NeurIPS 2019: 168-180

[C3] Some Constructions of Private, Efficient, and Optimal K-Norm and Elliptic Gaussian Noise. COLT 2024

[C4] Convergence of the EM Algorithm for Gaussian Mixtures with Unbalanced Mixing Coefficients. ICML 2012

[C5] Sparse Gaussian Conditional Random Fields: Algorithms, Theory, and Application to Energy Forecasting. ICML 2013

[C6] Classifying high-dimensional Gaussian mixtures: Where kernel methods fail and neural networks succeed. ICML 2021: 8936-8947

[C7] On the Properties of Kullback-Leibler Divergence Between Multivariate Gaussian Distributions. NeurIPS 2023

[C8] Precise Error Analysis of Regularized M-Estimators in High Dimensions. IEEE Trans. Inf. Theory, 2018

__For Question 7.__

[D1] Precise error analysis of regularized m-estimators in high dimensions. IEEE Trans. Inf. Theory. 2018.

[D2] The Noise-Sensitivity Phase Transition in Compressed Sensing. IEEE Trans. Inf. Theory. 2011

[D3] Positivity-preserving entropy stable schemes for the 1-D compressible Navier-Stokes equations: First-order approximation. J. Comput. Phys. 2022

[D4]. On Penalty Methods for Nonconvex Bilevel Optimization and First-Order Stochastic Approximation. ICLR 2024

[D5] Tracking MPC Tuning in Continuous Time: A First-Order Approximation of Economic MPC. IEEE Control. Syst. Lett. 2023

[D6] Sharp MSE Bounds for Proximal Denoising. Found. Comput. Math. 2016

---

### Decision · Program_Chairs · 2024-09-25

**Decision:**

Accept (poster)

**Comment:**

The spherically constrained least squares reformulation (SCLS) method proposed in a prior paper that won ICML 2022 Outstanding Paper Award is the main method for solving the Stackelberg prediction game with least squares loss. The current paper further enhances the theoretical framework of the SCLS method by providing a theoretical error analysis. The theoretical results indicate that the learner obtained through the SCLS method reliably converges to the actual learner vector of the Stackelberg prediction game. The reviewers were enthusiastic about this paper, nevertheless it feels that the current write-up can be improved in order to get a spotlight presentation (especially in terms of motivation), so the AC recommends acceptance as a poster.